# Characterizing the Exact Behaviors of Temporal Difference Learning Algorithms Using Markov Jump Linear System Theory

**Bin Hu,    Usman Ahmed Syed**
Department of Electrical and Computer Engineering
Coordinated Science Laboratory
University of Illinois at Urbana-Champaign

## Abstract

In this paper, we provide a unified analysis of temporal difference learning algorithms with linear function approximators by exploiting their connections to Markov jump linear systems (MJLS). We tailor the MJLS theory developed in the control community to characterize the exact behaviors of the first and second order moments of a large family of temporal difference learning algorithms. For both the IID and Markov noise cases, we show that the evolution of some augmented versions of the mean and covariance matrix of the TD estimation error exactly follows the trajectory of a deterministic linear time-invariant (LTI) dynamical system. Applying the well-known LTI system theory, we obtain closed-form expressions for the mean and covariance matrix of the TD estimation error at any time step. We provide a tight matrix spectral radius condition to guarantee the convergence of the covariance matrix of the TD estimation error, and perform a perturbation analysis to characterize the dependence of the TD behaviors on learning rate. For the IID case, we provide an exact formula characterizing how the mean and covariance matrix of the TD estimation error converge to the steady state values at a linear rate. For the Markov case, we use our formulas to explain how the behaviors of TD learning algorithms are affected by learning rate and the underlying Markov chain. For both cases, upper and lower bounds for the mean square TD error are derived. An exact formula for the steady state mean square TD error is also provided.

## 1 Introduction

Reinforcement learning (RL) has shown great promise in solving sequential decision making tasks [5, 48]. One important topic for RL is policy evaluation whose objective is to evaluate the value function of a given policy. A large family of temporal difference (TD) learning methods including standard TD, GTD, TDC, GTD2, DTD, and ATD [47, 50, 49, 38] have been developed to solve the policy evaluation problem. These TD learning algorithms have become important building blocks for RL algorithms. See [17] for a comprehensive survey. Despite the popularity of TD learning, the behaviors of these algorithms have not been fully understood from a theoretical viewpoint. The standard ODE technique [51, 9, 7, 36, 8] can only be used to prove asymptotic convergence. Finite sample bounds are challenging to obtain and typically developed in a case-by-case manner. Recently, there have been intensive research activities focusing on establishing finite sample bounds for TD learning methods with linear function approximations under various assumptions. The IID noise case is covered in [16, 37, 41]. In [6], the analysis is extended for a Markov noise model but an extra projection step in the algorithm is required. Very recently, finite sample bounds for the TD method (without the projection step) under the Markov assumption have been obtained in [45]. The bounds in [45] actually work for any TD learning algorithm that can be modeled by a linear

stochastic approximation scheme. It remains unclear how tight these bounds are (especially for the large learning rate region). To complement the existing analysis results and techniques, we propose a general unified analysis framework for TD learning algorithms by borrowing the Markov jump linear system (MJLS) theory [14] from the controls literature. Our approach is inspired by a recent research trend in applying control theory for analysis of optimization algorithms [39, 30, 31, 29, 21, 52, 15, 46, 28, 32, 22, 3, 40, 26, 4, 18, 43], and extends the jump system perspective for finite sum optimization methods in [31] to TD learning.

Our key insight is that TD learning algorithms with linear function approximations are essentially just Markov jump linear systems. Notice that a MJLS is described by a linear state space model whose state/input matrices are functions of a jump parameter sampled from a finite state Markov chain. Since the behaviors of MJLS have been well established in the controls field [14, 23, 1, 12, 13, 33, 34, 19, 20, 44], we can borrow the analysis tools there to analyze TD learning algorithms in a more unified manner. Our main contributions are summarized as follows.

1. We present a unified Markov jump linear system perspective on a large family of TD learning algorithms including TD, TDC, GTD, GTD2, ATD, and DTD. Specifically, we make the key observation that these methods are just MJLS subject to some prescribed input.

2. By tailoring the existing MJLS theory, we show that the evolution of some augmented versions of the mean and covariance matrix of the estimation error in all above TD learning methods exactly follows the trajectory of a deterministic linear time-invariant (LTI) dynamical system for both the IID and Markov noise cases. As a result, we obtain unified closed-form formulas for the mean and covariance matrix of the TD estimation error at any time step.

3. We provide a tight matrix spectral radius condition to guarantee the convergence of the covariance matrix of the TD estimation error under the general Markov assumption. By using the matrix perturbation theory [42, 35, 2, 24], we perform a perturbation analysis to show the dependence of the behaviors of TD learning on learning rate in a more transparent manner. For the IID case, we provide an exact formula characterizing how the mean and covariance matrix of the TD estimation error converge to the steady state values at a linear rate. For the Markov case, we use our formulas to explain how the behaviors of TD learning algorithms are affected by learning rate and the underlying Markov chain. For both cases, we have shown that the mean square error of TD learning converges linearly to a limit whose exact formula is also provided. In addition, upper and lower bounds for the mean square error of TD learning are simultaneously obtained.

We view our proposed analysis as a complement rather than a replacement for existing analysis techniques. The existing analysis focuses on upper bounds for the TD estimation error. Our closed-form formulas provide both upper and lower bounds for the mean square error of TD learning. Our analysis also characterizes the exact limit of the steady state TD error and related convergence rates.

## 2  Background

### 2.1  Notation

The set of $m$-dimensional real vectors is denoted as $\mathbb{R}^m$. The Kronecker product of two matrices $A$ and $B$ is denoted by $A \otimes B$. Notice $(A \otimes B)^T = A^T \otimes B^T$ and $(A \otimes B)(C \otimes D) = (AC) \otimes (BD)$ when the matrices have compatible dimensions. Let vec denote the standard vectorization operation that stacks the columns of a matrix into a vector. We have $\mathrm{vec}(AXB) = (B^\mathsf{T} \otimes A)\,\mathrm{vec}(X)$. Let sym denote the symmetrization operation, i.e. $\mathrm{sym}(A) = \frac{A^\mathsf{T}+A}{2}$. Let $\mathrm{diag}(H_i)$ denote a matrix whose $(i,i)$-th block is $H_i$ and all other blocks are zero. Specifically, given $H_i$ for $i = 1, \ldots, n$, we have

$$\mathrm{diag}(H_i) = \begin{bmatrix} H_1 & \ldots & 0 \\ \vdots & \ddots & \vdots \\ 0 & \ldots & H_n \end{bmatrix}.$$

A square matrix is Schur stable if all its eigenvalues have magnitude strictly less than 1. A square matrix is Hurwitz if all its eigenvalues have strictly negative real parts. The spectral radius of a matrix $H$ is denoted as $\sigma(H)$. The eigenvalue with the largest magnitude of $H$ is denoted as $\lambda_{\max}(H)$ and the eigenvalue with the largest real part of $H$ is denoted as $\lambda_{\max\,\mathrm{real}}(H)$.

## 2.2 Linear time-invariant systems

A linear time-invariant (LTI) system is typically governed by the following state-space model

$$x^{k+1} = \mathcal{H}x^k + \mathcal{G}u^k, \tag{1}$$

where $x^k \in \mathbb{R}^{n_x}$, $u^k \in \mathbb{R}^{n_u}$, $\mathcal{H} \in \mathbb{R}^{n_x \times n_x}$, and $\mathcal{G} \in \mathbb{R}^{n_x \times n_u}$. The LTI system theory has been well documented in standard control textbooks [27, 10]. Here we briefly review several useful results.

- **Closed-form formulas for $x^k$:** Given an initial condition $x^0$ and an input sequence $\{u^k\}$, the sequence $\{x^k\}$ can be determined using the following closed-form expression

$$x^k = (\mathcal{H})^k x^0 + \sum_{t=0}^{k-1} (\mathcal{H})^{k-1-t} \mathcal{G}u^t, \tag{2}$$

  where $(\mathcal{H})^k$ stands for the $k$-th power of the matrix $\mathcal{H}$.

- **Necessary and sufficient stability condition:** When $\mathcal{H}$ is Schur stable, we know $(\mathcal{H})^k x^0 \to 0$ for any arbitrary $x^0$. When $\sigma(\mathcal{H}) \geq 1$, there always exists $x^0$ such that $(\mathcal{H})^k x^0$ does not converge to 0. When $\sigma(\mathcal{H}) > 1$, there even exists $x^0$ such that $(\mathcal{H})^k x^0 \to \infty$. See Section 7.2 in [27] for a detailed discussion. A well-known result in the controls literature is that the LTI system (1) is stable if and only if $\mathcal{H}$ is Schur stable.

- **Exact limit for $x^k$:** If $\mathcal{H}$ is Schur stable and $u^k$ converges to a limit $u^\infty$, then $x^k$ will converge to an exact limit. This is formalized as follows.

  **Proposition 1.** *Consider the LTI system* (1). *If $\sigma(\mathcal{H}) < 1$ and $\lim_{k\to\infty} u^k = u^\infty$, then $\lim_{k\to\infty} x^k$ exists and we have $x^\infty = \lim_{k\to\infty} x^k = (I - \mathcal{H})^{-1} \mathcal{G}u^\infty$.*

- **Response for constant input:** If $u^k = u \; \forall k$ and $\sigma(\mathcal{H}) < 1$, then the closed-form expression for $x^k$ can be further simplified to give the following tight convergence rate result.

  **Proposition 2.** *Suppose $\sigma(\mathcal{H}) < 1$, and $x^k$ is determined by* (1). *If $u^k = u \; \forall k$, then $x^k$ converges to a limit point $x^\infty = \lim_{k\to\infty} x^k = (I - \mathcal{H})^{-1} \mathcal{G}u$. And we can compute $x^k$ as*

$$x^k = x^\infty + (\mathcal{H})^k (x^0 - x^\infty). \tag{3}$$

  *In addition, $\|x^k - x^\infty\| \leq C_0 (\sigma(\mathcal{H}) + \varepsilon)^k$ for some $C_0$ and any arbitrarily small $\varepsilon > 0$.*

  From the above proposition, we can clearly see that now $x^k$ is a sum of a constant steady state term $x^\infty$ and a matrix power term that decays at a linear rate specified by $\sigma(\mathcal{H})$ (see Section 2.2 in [39] for more explanations). The convergence rate characterized by $(\sigma(\mathcal{H}) + \varepsilon)$ is tight. More discussions on the tightness of this convergence rate are provided in the supplementary material.

- **Response for exponentially shrinking input:** When $u^k$ itself converges at a linear rate $\tilde{\rho}$ and $\mathcal{H}$ is Schur stable, $x^k$ will converge to its limit point at a linear rate specified by $\max\{\sigma(\mathcal{H}) + \varepsilon, \tilde{\rho}\}$. A formal statement is provided as follows.

  **Proposition 3.** *Suppose $\sigma(\mathcal{H}) < 1$, and $x^k$ is determined by* (1). *If $u^k$ converges to $u^\infty$ as $\|u^k - u^\infty\| \leq C\tilde{\rho}^k$, then we have $x^\infty = \lim_{k\to\infty} x^k = (I - \mathcal{H})^{-1} \mathcal{G}u^\infty$ and $\|x^k - x^\infty\| \leq C_0 \left(\max\{\sigma(\mathcal{H}) + \varepsilon, \tilde{\rho}\}\right)^k$ for some $C_0$ and any arbitrarily small $\varepsilon > 0$.*

The results in Propositions 1, 2, and 3 are well known in the control community. For completeness, we will include their proofs in the supplementary material.

## 2.3 Markov jump linear systems

Another important class of dynamic systems that have been extensively studied in the controls literature is the so-called Markov jump linear system (MJLS) [14]. Let $\{z^k\}$ be a Markov chain sampled from a finite state space $\mathcal{S}$. A MJLS is governed by the following state-space model:

$$\xi^{k+1} = H(z^k)\xi^k + G(z^k)y^k, \tag{4}$$

where $H(z^k)$ and $G(z^k)$ are matrix functions of $z^k$. Here, $\xi^k$ is the state, and $y^k$ is the input. There is a one-to-one mapping from $\mathcal{S}$ to the set $\mathcal{N} := \{1, 2, \dots, n\}$ where $n = |\mathcal{S}|$. We can assume $H(z^k)$ is sampled from a set of matrices $\{H_1, H_2, \dots, H_n\}$ and $G(z^k)$ is sampled from $\{G_1, G_2, \dots, G_n\}$. We have $H(z^k) = H_i$ and $G(z^k) = G_i$ when $z^k = i$. The MJLS theory has been well developed in the controls community [14]. We will apply the MJLS theory to analyze TD learning algorithms.

## 3 A general Markov jump system perspective for TD learning

In this section, we provide a general jump system perspective for TD learning with linear function approximations. Notice that many TD learning algorithms including TD, TDC, GTD, GTD2, A-TD, and D-TD can be modeled by the following linear stochastic recursion:

$$\xi^{k+1} = \xi^k + \alpha \left( A(z^k)\xi^k + b(z^k) \right), \tag{5}$$

where $\{z^k\}$ forms a finite state Markov chain and $b(z^k)$ satisfies $\lim_{k\to\infty} \mathbb{E}b(z^k) = 0$.[1] We have $A(z^k) = A_i$ and $b(z^k) = b_i$ when $z^k = i$. For simplicity, we mainly focus on analyzing (5). Other models including two time-scale schemes [25, 54] will be discussed in the supplementary material.

Our key observation is that (5) can be rewritten as the following MJLS

$$\xi^{k+1} = (I + \alpha A(z^k))\xi^k + \alpha b(z^k). \tag{6}$$

The above model is a special case of (4) if we set $H(z^k) = I + \alpha A(z^k)$, $G(z^k) = \alpha b(z^k)$, and $y^k = 1 \,\forall k$. Consequently, many TD learning algorithms can be analyzed using the MJLS theory.

We will borrow the analysis idea from the standard MJLS theory. Our analysis is built upon the fact that some augmented versions of the mean and the covariance matrix of $\{\xi^k\}$ for the MJLS model (4) actually follow the dynamics of a deterministic LTI model in the form of (1) [14, Chapter 3]. To see this, we denote the transition probabilities for the Markov chain $\{z^k\}$ as $p_{ij} := \mathbb{P}(z^{k+1} = j | z^k = i)$ and specify the transition matrix $P$ by setting its $(i, j)$-th entry to be $p_{ij}$. Obviously, we have $p_{ij} \geq 0$ and $\sum_{j=1}^n p_{ij} = 1$ for all $i$. Next, the indicator function $\mathbf{1}_{\{z^k=i\}}$ is defined as $\mathbf{1}_{\{z^k=i\}} = 1$ if $z^k = i$ and $\mathbf{1}_{\{z^k=i\}} = 0$ otherwise. Now we define the following key quantities:

$$q_i^k = \mathbb{E}\left( \xi^k \mathbf{1}_{\{z^k=i\}} \right), \quad Q_i^k = \mathbb{E}\left( \xi^k (\xi^k)^\mathsf{T} \mathbf{1}_{\{z^k=i\}} \right).$$

Suppose $y^k = 1 \,\forall k$. Based on [14, Proposition 3.35], $q^k$ and $Q^k$ can be iteratively calculated as

$$q_j^{k+1} = \sum_{i=1}^n p_{ij}(H_i q_i^k + G_i p_i^k), \tag{7}$$

$$Q_j^{k+1} = \sum_{i=1}^n p_{ij} \left( H_i Q_i^k H_i^\mathsf{T} + 2\operatorname{sym}(H_i q_i^k G_i^\mathsf{T}) + p_i^k G_i G_i^\mathsf{T} \right), \tag{8}$$

where $p_i^k := \mathbb{P}(z^k = i)$. If we further augment $q_i^k$ and $Q_i^k$ as

$$q^k = \begin{bmatrix} q_1^k \\ \vdots \\ q_n^k \end{bmatrix}, \quad Q^k = \begin{bmatrix} Q_1^k & Q_2^k & \cdots & Q_n^k \end{bmatrix},$$

then it is straightforward to rewrite (7) (8) as the following LTI system

$$\begin{bmatrix} q^{k+1} \\ \operatorname{vec}(Q^{k+1}) \end{bmatrix} = \begin{bmatrix} \mathcal{H}_{11} & 0 \\ \mathcal{H}_{21} & \mathcal{H}_{22} \end{bmatrix} \begin{bmatrix} q^k \\ \operatorname{vec}(Q^k) \end{bmatrix} + \begin{bmatrix} u_q^k \\ u_Q^k \end{bmatrix}, \tag{9}$$

where $\mathcal{H}_{11}, \mathcal{H}_{21}, \mathcal{H}_{22}, u_q^k$, and $u_Q^k$ are given by

$$\mathcal{H}_{11} = \begin{bmatrix} p_{11}H_1 & \cdots & p_{n1}H_n \\ \vdots & \ddots & \vdots \\ p_{1n}H_1 & \cdots & p_{nn}H_n \end{bmatrix}, \mathcal{H}_{22} = \begin{bmatrix} p_{11}H_1 \otimes H_1 & \cdots & p_{n1}H_n \otimes H_n \\ \vdots & \ddots & \vdots \\ p_{1n}H_1 \otimes H_1 & \cdots & p_{nn}H_n \otimes H_n \end{bmatrix},$$

$$\mathcal{H}_{21} = \begin{bmatrix} p_{11}(H_1 \otimes G_1 + G_1 \otimes H_1) & \cdots & p_{n1}(H_n \otimes G_n + G_n \otimes H_n), \\ \vdots & \ddots & \vdots \\ p_{1n}(H_1 \otimes G_1 + G_1 \otimes H_1) & \cdots & p_{nn}(H_n \otimes G_n + G_n \otimes H_n) \end{bmatrix},$$

$$u_q^k = \begin{bmatrix} p_{11}G_1 & \cdots & p_{n1}G_n \\ \vdots & \ddots & \vdots \\ p_{1n}G_1 & \cdots & p_{nn}G_n \end{bmatrix} \begin{bmatrix} p_1^k I \\ \vdots \\ p_n^k I \end{bmatrix}, u_Q^k = \begin{bmatrix} p_{11}G_1 \otimes G_1 & \cdots & p_{n1}G_n \otimes G_n \\ \vdots & \ddots & \vdots \\ p_{1n}G_1 \otimes G_1 & \cdots & p_{nn}G_n \otimes G_n \end{bmatrix} \begin{bmatrix} p_1^k I \\ \vdots \\ p_n^k I \end{bmatrix}. \tag{10}$$

A detailed derivation for the above result is presented in the supplementary material. A key implication here is that $q^k$ and $\text{vec}(Q^k)$ follow the LTI dynamics (9) and can be analyzed using the standard LTI theory reviewed in Section 2.2. Obviously, we have $\mathbb{E}\xi^k = \sum_{i=1}^n q_i^k$, $\mathbb{E}\left(\xi^k(\xi^k)^\mathsf{T}\right) = \sum_{i=1}^n Q_i^k$, and $\mathbb{E}\|\xi^k\|^2 = \text{trace}(\sum_{i=1}^n Q_i^k) = (\mathbf{1}_n^\mathsf{T} \otimes \text{vec}(I_{n_\xi})^\mathsf{T})\text{vec}(Q^k)$. Hence the mean, covariance, and mean square norm of $\xi^k$ can all be calculated using closed-form expressions. We will present a detailed analysis for (6) and provide related implications for TD learning in the next two sections.

For illustrative purposes, we explain the jump system perspective for the standard TD method.

**Example 1: TD method.** The standard TD method (or TD(0)) uses the following update rule:

$$\theta^{k+1} = \theta^k - \alpha\phi(s^k)\left((\phi(s^k) - \gamma\phi(s^{k+1}))^\mathsf{T}\theta^k - r(s^k)\right), \tag{11}$$

where $\{s^k\}$ is the underlying Markov chain, $\phi$ is the feature vector, $r$ is the reward, $\gamma$ is the discounting factor, and $\theta^k$ is the weight vector to be estimated. Suppose $\theta^*$ is the vector that solves the projected Bellman equation. We can set $z^k = \left[(s^{k+1})^\mathsf{T} \quad (s^k)^\mathsf{T}\right]^\mathsf{T}$ and then rewrite the TD update as

$$\theta^{k+1} - \theta^* = \left(I + \alpha A(z^k)\right)(\theta^k - \theta^*) + \alpha b(z^k), \tag{12}$$

where $A(z^k) = \phi(s^k)(\gamma\phi(s^{k+1}) - \phi(s^k))^\mathsf{T}$ and $b(z^k) = \phi(s^k)\left(r(s^k) + (\phi(s^k) - \gamma\phi(s^{k+1}))^\mathsf{T}\theta^*\right)$. Suppose $\lim_{k\to\infty} p_i^k = p_i^\infty$. Since the projected Bellman equation and the equation $\sum_{i=1}^n p_i^\infty b_i = 0$ are actually equivalent, we have naturally enforced $\lim_{k\to\infty}\mathbb{E}b(z^k) = 0$. Therefore, the TD update can be modeled as (6) with $b(z^k)$ satisfying $\lim_{k\to\infty}\mathbb{E}b(z^k) = 0$. See Section 3.1 in [45] for a similar formulation. Now we can apply the MJLS theory and the LTI model (9) to analyze the covariance $\mathbb{E}\left((\theta^k - \theta^*)(\theta^k - \theta^*)^\mathsf{T}\right)$ and the mean square error $\mathbb{E}\|\theta^k - \theta^*\|^2$. In this case, we have

$$q^k = \begin{bmatrix} \mathbb{E}\left((\theta^k - \theta^*)\mathbf{1}_{\{z^k=1\}}\right) \\ \vdots \\ \mathbb{E}\left((\theta^k - \theta^*)\mathbf{1}_{\{z^k=n\}}\right) \end{bmatrix}, \quad \text{vec}(Q^k) = \begin{bmatrix} \text{vec}\left(\mathbb{E}((\theta^k - \theta^*)(\theta^k - \theta^*)^\mathsf{T}\mathbf{1}_{\{z^k=1\}})\right) \\ \vdots \\ \text{vec}\left(\mathbb{E}((\theta^k - \theta^*)(\theta^k - \theta^*)^\mathsf{T}\mathbf{1}_{\{z^k=n\}})\right) \end{bmatrix}.$$

Then we can easily analyze $q^k$ and $Q^k$ by applying the LTI model (9). In general, the covariance matrix $\mathbb{E}\left((\theta^k - \theta^*)(\theta^k - \theta^*)^\mathsf{T}\right)$ and the mean value $\mathbb{E}(\theta^k - \theta^*)$ do not directly follow an LTI system. However, when working with the augmented covariance matrix $Q^k$ and the augmented mean value vector $q^k$, we do obtain an LTI model in the form of (1). Once the closed-form expression for $Q^k$ is obtained, the mean square estimation error for the TD update can be immediately calculated as $\mathbb{E}\|\theta^k - \theta^*\|^2 = \text{trace}(\sum_{i=1}^n Q_i^k) = (\mathbf{1}_n^\mathsf{T} \otimes \text{vec}(I_{n_\theta})^\mathsf{T})\text{vec}(Q^k)$.

Here we omit the detailed formulations for other TD learning methods. The key message is that $\{z^k\}$ can be viewed as a jump parameter and TD learning methods are essentially just MJLS. Notice that all the TD learning algorithms that can be analyzed using the ODE method are in the form of (6). Jump system perspectives for other TD learning algorithms are discussed in the supplementary material.

**Remark 1** (Assumptions). *Denote $\bar{A} = \lim_{k\to\infty}\mathbb{E}A(z^k) = \sum_{i=1}^n p_i^\infty A_i$. In this paper, we will assume $\bar{A}$ is Hurwitz. This assumption is standard and even required by the ODE approach. For the standard TD method, $\bar{A}$ is Hurwitz when the discount factor is smaller than 1, $p_i^\infty$ is positive for all $i$, and the feature matrix is full column rank [51]. It is worth emphasizing that the assumption $\lim_{k\to\infty}\mathbb{E}b(z^k) = 0$ is also general. Suppose $\sum_{i=1}^n p_i^\infty b_i \neq 0$. This case can still be handled using a shifting argument since $\bar{A}$ is Hurwitz. Notice the iteration $\xi^{k+1} = (I + \alpha A(z^k))\xi^k + \alpha b(z^k)$ can be rewritten as $\xi^{k+1} - \tilde{\xi} = \xi^k - \tilde{\xi} + \alpha\left(A(z^k)(\xi^k - \tilde{\xi}) + A(z^k)\tilde{\xi} + b(z^k)\right)$ for any $\tilde{\xi}$. Now we denote $\tilde{b}_i = A_i\tilde{\xi} + b_i$ and the above iteration just becomes $\xi^{k+1} - \tilde{\xi} = (I + \alpha A(z^k))(\xi^k - \tilde{\xi}) + \alpha\tilde{b}(z^k)$. When $\bar{A}$ is Hurwitz (and hence invertible), we can choose $\tilde{\xi} = -(\sum_{i=1}^n p_i^\infty A_i)^{-1}(\sum_{i=1}^n p_i^\infty b_i)$ such that $\sum_{i=1}^n p_i^\infty \tilde{b}_i = \sum_{i=1}^n p_i^\infty(A_i\tilde{\xi} + b_i) = 0$.*

**Remark 2** (Generality of (4)). *Notice that (4) provides a general jump system model for linear stochastic schemes that may have more complicated forms than (5). However, (4) can not be directly used to cover nonlinear stochastic approximation schemes. See [53, 11] for recent finite sample analysis results on nonlinear stochastic approximation over non-IID data.*

## 4    Analysis under the IID assumption

For illustrative purposes, we first present the analysis for (6) under the IID assumption ($\mathbb{P}(z^k = i) = p_i \ \forall i$). In this case, the analysis is significantly simpler, since $\{\mathbb{E}\xi^k\}$ and $\{\mathbb{E}\left(\xi^k(\xi^k)^\mathsf{T}\right)\}$ directly form LTI systems with much smaller dimensions. We denote $\mu^k := \mathbb{E}\xi^k$ and $\mathbb{Q}^k := \mathbb{E}\left(\xi^k(\xi^k)^\mathsf{T}\right)$. Then the following equations hold for the general jump system model (4)

$$\mu^{k+1} = \sum_{i=1}^{n} p_i(H_i\mu^k + G_i) = \bar{H}\mu^k + \bar{G},$$

$$\mathrm{vec}(\mathbb{Q}^{k+1}) = \left(\sum_{i=1}^{n} p_i H_i \otimes H_i\right)\mathrm{vec}(\mathbb{Q}^k) + \left(\sum_{i=1}^{n} p_i(H_i \otimes G_i + G_i \otimes H_i)\right)\mu^k + \sum_{i=1}^{n} p_i G_i \otimes G_i.$$
(13)

There are many ways to derive the above formulas. One way is to first show $q_i^k = p_i\mu^k$ and $Q_i^k = p_i\mathbb{Q}^k$ in this case and then apply (7) and (8). Another way is to directly modify the proof of Theorem 1 (which is presented in the supplementary material). Now consider the jump system model (6) under the assumption $\mathbb{E}b(z^k) = \sum_{i=1}^{n} p_i b_i = 0$. In this case, we have $H_i = I + \alpha A_i$, $G_i = \alpha b_i$, and $y^k = 1$. Denote $\bar{A} := \sum_{i=1}^{n} p_i A_i$. We can directly obtain the following result.

**Theorem 1.** *Consider the jump system model* (6) *with* $H_i = I + \alpha A_i$, $G_i = \alpha b_i$, *and* $y^k = 1$. *Suppose* $\{z^k\}$ *is sampled from* $\mathcal{N}$ *using an IID distribution* $\mathbb{P}(z^k = i) = p_i$. *In addition, assume* $\sum_{i=1}^{n} p_i b_i = 0$. *Then* $\mu^k$ *and* $\mathrm{vec}(\mathbb{Q}^k)$ *are governed by the following LTI system:*

$$\begin{bmatrix} \mu^{k+1} \\ \mathrm{vec}(\mathbb{Q}^{k+1}) \end{bmatrix} = \begin{bmatrix} \mathcal{H}_{11} & 0 \\ \mathcal{H}_{21} & \mathcal{H}_{22} \end{bmatrix} \begin{bmatrix} \mu^k \\ \mathrm{vec}(\mathbb{Q}^k) \end{bmatrix} + \begin{bmatrix} 0 \\ \alpha^2 \sum_{i=1}^{n} p_i(b_i \otimes b_i) \end{bmatrix},$$
(14)

*where* $\mathcal{H}_{11}$, $\mathcal{H}_{21}$ *and* $\mathcal{H}_{22}$ *are determined as*

$$\mathcal{H}_{11} = I + \alpha\bar{A},$$

$$\mathcal{H}_{21} = \alpha^2 \sum_{i=1}^{n} p_i(A_i \otimes b_i + b_i \otimes A_i),$$
(15)

$$\mathcal{H}_{22} = I_{n_\xi^2} + \alpha(I \otimes \bar{A} + \bar{A} \otimes I) + \alpha^2 \sum_{i=1}^{n} p_i(A_i \otimes A_i).$$

*In addition, if* $\sigma(\mathcal{H}_{22}) < 1$, *we have*

$$\begin{bmatrix} \mu^k \\ \mathrm{vec}(\mathbb{Q}^k) \end{bmatrix} = \left(\begin{bmatrix} \mathcal{H}_{11} & 0 \\ \mathcal{H}_{21} & \mathcal{H}_{22} \end{bmatrix}\right)^k \left(\begin{bmatrix} \mu^0 \\ \mathrm{vec}(\mathbb{Q}^0) \end{bmatrix} - \begin{bmatrix} \mu^\infty \\ \mathrm{vec}(\mathbb{Q}^\infty) \end{bmatrix}\right) + \begin{bmatrix} \mu^\infty \\ \mathrm{vec}(\mathbb{Q}^\infty) \end{bmatrix}$$
(16)

*where* $\mu^\infty = \lim_{k\to\infty} \mu^k = 0$, *and* $\mathrm{vec}(\mathbb{Q}^\infty)$ *is given as*

$$\mathrm{vec}(\mathbb{Q}^\infty) = \lim_{k\to 0} \mathrm{vec}(\mathbb{Q}^k) = -\alpha\left(I \otimes \bar{A} + \bar{A} \otimes I + \alpha \sum_{i=1}^{n} p_i(A_i \otimes A_i)\right)^{-1}\left(\sum_{i=1}^{n} p_i(b_i \otimes b_i)\right)$$
(17)

*Proof.* For completeness, a detailed proof is presented in the supplementary material.    □

Now we discuss the implications of the above theorem for TD learning. For simplicity, we denote $\mathcal{H} = \begin{bmatrix} \mathcal{H}_{11} & 0 \\ \mathcal{H}_{21} & \mathcal{H}_{22} \end{bmatrix}$.

**Stability condition for TD learning.**    From the LTI theory, the system (14) is stable if and only if $\mathcal{H}$ is Schur stable. We can apply Proposition 3.6 in [14] to show that $\mathcal{H}$ is Schur stable if and only if $\mathcal{H}_{22}$ is Schur stable. Hence, a necessary and sufficient stability condition for the LTI system (14) is that $\mathcal{H}_{22}$ is Schur stable. Under this condition, the first term on the right side of (16) converges to 0 at a linear rate specified by $\sigma(\mathcal{H})$, and the second term on the right side of (16) is a constant

matrix quantifying the steady state covariance. An important question for TD learning is how to choose $\alpha$ such that $\sigma(\mathcal{H}_{22}) < 1$ for some given $\{A_i\}$, $\{b_i\}$, and $\{p_i\}$. We provide some clue to this question by applying an eigenvalue perturbation analysis to the matrix $\mathcal{H}_{22}$. We assume $\alpha$ is small. Then under mild technical condition[2], we can ignore the quadratic term $\alpha^2 \sum_{i=1}^{n} p_i(A_i \otimes A_i)$ in the expression of $\mathcal{H}_{22}$ and use $\lambda_{\max}(I_{n_{\xi}^2} + \alpha(I \otimes \bar{A} + \bar{A} \otimes I))$ to estimate $\lambda_{\max}(\mathcal{H}_{22})$. We have

$$\lambda_{\max}(\mathcal{H}_{22}) = 1 + 2\lambda_{\max \, \text{real}}(\bar{A})\alpha + O(\alpha^2). \tag{18}$$

Then we immediately obtain $\sigma(\mathcal{H}_{22}) \approx 1 + 2\,\text{real}(\lambda_{\max \, \text{real}}(\bar{A}))\alpha + O(\alpha^2)$. Therefore, as long as $\bar{A}$ is Hurwitz, there exists sufficiently small $\alpha$ such that $\sigma(\mathcal{H}_{22}) < 1$. More details of the perturbation analysis are provided in the supplementary material.

**Exact limit for the mean square error of TD learning.** Obviously, $\mu^k$ converges to 0 at the rate specified by $\sigma(I + \alpha\bar{A})$ due to the relation $\mu^k = (I + \alpha\bar{A})^k \mu^0$. Applying Proposition 3 and making use of the block structure in $\mathcal{H}$, one can show $\text{vec}(\mathbb{Q}^{\infty}) = \alpha^2(I_{n_{\xi}^2} - \mathcal{H}_{22})^{-1} \left( \sum_{i=1}^{n} p_i(b_i \otimes b_i) \right)$, which leads to the result in (17). A key message here is that the covariance matrix converges linearly to an exact limit under the stability condition $\sigma(\mathcal{H}_{22}) < 1$. We can clearly see $\lim_{k \to 0} \text{vec}(\mathbb{Q}^k) = O(\alpha)$ and can be controlled by decreasing $\alpha$. When $\alpha$ is large, we need to keep the quadratic term $\alpha \sum_{i=1}^{n} p_i(A_i \otimes A_i)$. Therefore, our theory captures the steady-state behavior of TD learning for both small and large $\alpha$, and complement the existing finite sample bounds in literatures. To further compare our results with existing finite sample bounds, we obtain the following result for the mean square error of TD learning.

**Corollary 1.** *Consider the TD update* (12) *with $\bar{A}$ being Hurwitz. Suppose $\sigma(\mathcal{H}_{22}) < 1$ and $\mathbb{P}(z^k = i) = p_i \, \forall i$. Then $\lim_{k \to \infty} \mathbb{E}\|\theta^k - \theta^*\|^2$ exists and is determined as $\delta^{\infty} := \lim_{k \to \infty} \mathbb{E}\|\theta^k - \theta^*\|^2 = \text{trace}(\mathbb{Q}^{\infty})$ where $\mathbb{Q}^{\infty}$ is given by* (17). *In addition, the following mean square TD error bounds hold for some constant $C_0$ and any arbitrary small positive $\varepsilon$:*

$$\delta^{\infty} - C_0(\sigma(\mathcal{H}) + \varepsilon)^k \leq \mathbb{E}\|\theta^k - \theta^*\|^2 \leq \delta^{\infty} + C_0(\sigma(\mathcal{H}) + \varepsilon)^k. \tag{19}$$

*Finally, for sufficiently small $\alpha$, one has $\lim_{k \to \infty} \mathbb{E}\|\theta^k - \theta^*\|^2 = O(\alpha)$. If $\lambda_{\max}(I_{n_{\xi}^2} + \alpha(I \otimes \bar{A} + \bar{A} \otimes I))$ is a semisimple eigenvalue, then $\sigma(\mathcal{H}) = \sigma(\mathcal{H}_{11}) = 1 + \text{real}(\lambda_{\max \, \text{real}}(\bar{A}))\alpha$ for small $\alpha$.*

*Proof.* Recall that we have $\mathbb{E}\|\theta^k - \theta^*\|^2 = \text{trace}(\mathbb{Q}^k)$. Taking limits on both sides leads to the expression for $\delta^{\infty}$. Then we can apply Proposition 2 to obtain a linear convergence bound for $\mathbb{Q}^k$ which eventually leads to (19). Notice $\bar{A}$ is assumed to be Hurwitz. Therefore, we can apply standard matrix perturbation theory to show $\delta^{\infty} = O(\alpha)$ and $\sigma(\mathcal{H}) = \sigma(\mathcal{H}_{11}) = 1 + \text{real}(\lambda_{\max \, \text{real}}(\bar{A}))\alpha$ for sufficiently small $\alpha$. □

The above corollary gives both upper and lower bounds for the mean square error of TD learning. From the above result, the final TD estimation error is actually exactly on the order of $O(\alpha)$. This justifies the tightness of the existing upper bounds for the final TD error up to a constant factor. From the above corollary, we can also see that one can obtain a faster convergence rate at the price of getting a bigger steady state error. This is consistent with the finite sample bound in the literature [6, 45]. Since $\mathcal{H}_{21} = O(\alpha^2)$, it is possible to tighten the rate as $\sigma(\mathcal{H}_{22}) \approx 1 + 2\,\text{real}(\lambda_{\max \, \text{real}}(\bar{A}))\alpha$ by allowing some extra error on the order of $O(\alpha)$. We omit the details for such modifications.

## 5 Analysis under the Markov assumption

Now we analyze the behaviors of TD learning under the general assumption that $\{z^k\}$ is a Markov chain. Recall that the augmented mean vector $q^k$ and the augmented covariance matrix $Q^k$ have been defined in Section 3. We can directly modify (9) to obtain the following result.

**Theorem 2.** *Consider the jump system model* (6) *with $H_i = I + \alpha A_i$, $G_i = \alpha b_i$, and $y^k = 1$. Suppose $\{z^k\}$ is a Markov chain sampled from $\mathcal{N}$ using the transition matrix $P$. In addition, define $p_i^k = \mathbb{P}(z^k = i)$ and set the augmented vector $p^k := \begin{bmatrix} p_1^k & p_2^k & \dots & p_n^k \end{bmatrix}^{\mathsf{T}}$. Clearly $p^k = (P^{\mathsf{T}})^k p^0$. Further denote the augmented vectors as $b := \begin{bmatrix} b_1^{\mathsf{T}} & b_2^{\mathsf{T}} & \dots & b_n^{\mathsf{T}} \end{bmatrix}^{\mathsf{T}}$,*

$\hat{B} = \left[(b_1 \otimes b_1)^{\mathsf{T}} \quad \ldots \quad (b_n \otimes b_n)^{\mathsf{T}}\right]^{\mathsf{T}}$, *and set* $S(b_i, A_i) := (b_i \otimes (I + \alpha A_i) + (I + \alpha A_i) \otimes b_i)$.
*Then $q^k$ and $\mathrm{vec}(Q^k)$ are governed by the following LTI model:*

$$\begin{bmatrix} q^{k+1} \\ \mathrm{vec}(Q^{k+1}) \end{bmatrix} = \begin{bmatrix} \mathcal{H}_{11} & 0 \\ \mathcal{H}_{21} & \mathcal{H}_{22} \end{bmatrix} \begin{bmatrix} q^k \\ \mathrm{vec}(Q^k) \end{bmatrix} + \begin{bmatrix} \alpha((P^{\mathsf{T}} \mathrm{diag}(p_i^k)) \otimes I_{n_\xi})b \\ \alpha^2((P^{\mathsf{T}} \mathrm{diag}(p_i^k)) \otimes I_{n_\xi^2})\hat{B} \end{bmatrix}, \tag{20}$$

*where $\mathcal{H}_{11}$, $\mathcal{H}_{21}$ and $\mathcal{H}_{22}$ are given by*

$$\mathcal{H}_{11} = (P^{\mathsf{T}} \otimes I_{n_\xi}) \mathrm{diag}(I_{n_\xi} + \alpha A_i),$$

$$\mathcal{H}_{21} = \alpha \begin{bmatrix} p_{11}S(b_1, A_1) & \ldots & p_{n1}S(b_n, A_n) \\ \vdots & \ddots & \vdots \\ p_{1n}S(b_1, A_1) & \ldots & p_{nn}S(b_n, A_n) \end{bmatrix}, \tag{21}$$

$$\mathcal{H}_{22} = (P^{\mathsf{T}} \otimes I_{n_\xi^2}) \mathrm{diag}((I_{n_\xi} + \alpha A_i) \otimes (I_{n_\xi} + \alpha A_i)).$$

*In addition, the following closed-form solution holds for any $k$*

$$q^k = (\mathcal{H}_{11})^k q^0 + \alpha \sum_{t=0}^{k-1} (\mathcal{H}_{11})^{k-1-t}((P^{\mathsf{T}} \mathrm{diag}(p_i^t)) \otimes I_{n_\xi})b,$$

$$\mathrm{vec}(Q^k) = (\mathcal{H}_{22})^k \mathrm{vec}(Q^0) + \sum_{t=0}^{k-1} (\mathcal{H}_{22})^{k-1-t} \left( \mathcal{H}_{21}q^t + \alpha^2((P^{\mathsf{T}} \mathrm{diag}(p_i^t)) \otimes I_{n_\xi^2})\hat{B} \right), \tag{22}$$

*where $\mathcal{H}_{11}$, $\mathcal{H}_{21}$ and $\mathcal{H}_{22}$ are determined by (21).*

*Proof.* A detailed proof is presented in the supplementary material. We present a proof sketch here. Notice (20) is a direct consequence of (7) and (8) (which are special cases of Proposition 3.35 in [14]). Specifically, it is straightforward to verify the following equations using the Markov assumption

$$q_j^{k+1} = \sum_{i=1}^n p_{ij} \left( (I + \alpha A_i)q_i^k + \alpha p_i^k b_i \right), \tag{23}$$

$$Q_j^{k+1} = \sum_{i=1}^n p_{ij} \left( (I + \alpha A_i)Q_i^k(I + \alpha A_i)^{\mathsf{T}} + 2\alpha \, \mathrm{sym}((I + \alpha A_i)q_i^k b_i^{\mathsf{T}}) + \alpha^2 p_i^k b_i b_i^{\mathsf{T}} \right). \tag{24}$$

Then we can apply the basic property of the vectorization operation to obtain (20). Applying (2) to iterate (20) directly leads to (22). $\qquad\square$

Therefore, the evolutions of $q^k$ and $Q^k$ can be fully understood via the well-established LTI system theory. Now we discuss the implications of Theorem 2 for TD learning.

**Stability condition for TD learning.** Similar to the IID case, the necessary and sufficient stability condition is $\sigma(\mathcal{H}_{22}) < 1$. Now $\mathcal{H}_{22}$ becomes a much larger matrix depending on the transition matrix $P$. An important question is how to choose $\alpha$ such that $\sigma(\mathcal{H}_{22}) < 1$ for some given $\{A_i\}$, $\{b_i\}$, $P$, and $\{p^0\}$. Again, we perform an eigenvalue perturbation analysis for the matrix $\mathcal{H}_{22}$. This case is quite subtle due to the fact that we are no longer perturbing an identity matrix. We are perturbing the matrix $(P^{\mathsf{T}} \otimes I_{n_\xi^2})$ and the eigenvalues here are not simple. Under the ergodicity assumption, the largest eigenvalue for $(P^{\mathsf{T}} \otimes I_{n_\xi^2})$ (which is 1) is semisimple. Hence we can directly apply the results in Section II of [35] or Theorem 2.1 in [42] to show

$$\lambda_{\max}(\mathcal{H}_{22}) = 1 + 2\lambda_{\max \, \mathrm{real}}(\bar{A})\alpha + o(\alpha), \tag{25}$$

where $\bar{A} = \sum_{i=1}^n p_i^\infty A_i$ and $p^\infty$ is the unique stationary distribution of the Markov chain under the ergodicity assumption. Then we still have $\sigma(\mathcal{H}_{22}) \approx 1 + 2\,\mathrm{real}(\lambda_{\max \, \mathrm{real}}(\bar{A}))\alpha + o(\alpha)$. Therefore, as long as $\bar{A}$ is Hurwitz, there exists sufficiently small $\alpha$ such that $\sigma(\mathcal{H}_{22}) < 1$. This is consistent with Assumption 3 in [45]. To understand the details of our perturbation argument, we refer the readers to the remark placed after Theorem 2.1 in [42]. Notice we have

$$\mathcal{H}_{22} = P^{\mathsf{T}} \otimes I_{n_\xi^2} + \alpha(P^{\mathsf{T}} \otimes I_{n_\xi^2}) \mathrm{diag}(A_i \otimes I + I \otimes A_i) + O(\alpha^2).$$

The largest eigenvalue of $P^{\mathsf{T}} \otimes I_{n_\xi^2}$ is semisimple due to the ergodicity assumption. Then the perturbation result directly follows as a consequence of Theorem 2.1 in [42]. More explanations are also provided in the supplementary material.

**Exact limit for the mean square TD error and related convergence rate.** Assume the Markov chain $\{z^k\}$ is aperiodic and irreducible. Then we have $p^t \to p^\infty$ at some linear rate where $p^\infty$ is the stationary distribution. In this case, we can apply Proposition 3 to show that the mean square error of TD learning converges linearly to an exact limit.

**Corollary 2.** *Consider the TD update* (12) *with $\bar{A}$ being Hurwitz. Let $\{z^k\}$ be a Markov chain sampled from $\mathcal{N}$ using the transition matrix $P$. Suppose $\sigma(\mathcal{H}_{22}) < 1$. We set $N = nn_\xi^2$. If we assume $p^k \to p^\infty$ where $p^\infty$ is the stationary distribution for $\{z^k\}$, then we have*

$$q^\infty = \lim_{k\to\infty} q^k = \alpha(I - \mathcal{H}_{11})^{-1}((P^\mathsf{T} \operatorname{diag}(p_i^\infty)) \otimes I_{n_\xi})b,$$

$$\operatorname{vec}(Q^\infty) = \lim_{k\to 0} \operatorname{vec}(Q^k) = \alpha^2 (I_N - \mathcal{H}_{22})^{-1} \left( \alpha^{-2} \mathcal{H}_{21} q^\infty + ((P^\mathsf{T} \operatorname{diag}(p_i^\infty)) \otimes I_{n_\xi^2}) \hat{B} \right), \quad (26)$$

$$\delta^\infty = \lim_{k\to\infty} \mathbb{E}\|\theta^k - \theta^*\|^2 = (\mathbf{1}_n^\mathsf{T} \otimes \operatorname{vec}(I_{n_\theta})^\mathsf{T}) \operatorname{vec}(Q^\infty).$$

*If we further assume the geometric ergodicity, i.e. $\|p^k - p^\infty\| \le C\tilde{\rho}^k$, then we have*

$$\delta^\infty - C_0 \max\{\sigma(\mathcal{H}) + \varepsilon, \tilde{\rho}\}^k \le \mathbb{E}\|\theta^k - \theta^*\|^2 \le \delta^\infty + C_0 \max\{\sigma(\mathcal{H}) + \varepsilon, \tilde{\rho}\}^k, \quad (27)$$

*where $C_0$ is some constant and $\varepsilon$ is an arbitrary small positive number. For sufficiently small $\alpha$, we have $\delta^\infty = O(\alpha)$. If $\lambda_{\max}(P^\mathsf{T} \otimes I_{n_\xi^2} + \alpha(P^\mathsf{T} \otimes I_{n_\xi^2}) \operatorname{diag}(A_i \otimes I + I \otimes A_i))$ is a semisimple eigenvalue, then we further have $\sigma(\mathcal{H}) = 1 + \operatorname{real}(\lambda_{\max\,\mathrm{real}}(\bar{A}))\alpha + o(\alpha)$ for sufficiently small $\alpha$.*

*Proof.* Notice $\mathbb{E}\|\theta^k - \theta^*\|^2 = (\mathbf{1}_n^\mathsf{T} \otimes \operatorname{vec}(I_{n_\xi})^\mathsf{T}) \operatorname{vec}(Q^k)$. We can directly apply Theorem 2, Proposition 1, and Proposition 3 to prove (26) and (27). When $\alpha$ is small, we can apply the Laurent series trick in [2, 24] to show that $\lim_{k\to 0} \operatorname{vec}(Q^k) = O(\alpha)$ and $\delta^\infty = O(\alpha)$. The difficulty here is that $I_N - P^\mathsf{T} \otimes I_{n_\xi^2}$ is a singular matrix and hence $(I_N - \mathcal{H}_{22})^{-1}$ does not have a Taylor series around $\alpha = 0$. Therefore, we need to apply some advanced matrix inverse perturbation result to perform a Laurent expansion of $(I_N - \mathcal{H}_{22})^{-1}$. By using the ergodicity assumption and the matrix inverse perturbation theory in [2, 24], we can obtain the Laurent expansion of $(I_N - \mathcal{H}_{22})^{-1}$ and show $\lim_{k\to 0} \operatorname{vec}(Q^k) = O(\alpha)$. Consequently, we have $\delta^\infty = O(\alpha)$. By applying Theorem 2.1 in [42], we can show $\sigma(\mathcal{H}) = 1 + \operatorname{real}(\lambda_{\max\,\mathrm{real}}(\bar{A}))\alpha + o(\alpha)$. $\qquad\square$

Due to the assumption $\sum_{i=1}^n p_i^\infty b_i = 0$, we have $\lim_{k\to\infty} q^k \ne 0$ in general but $\mu^\infty = 0$. Again, we have obtained both upper and lower bounds for the mean square TD error. Our result states that under mild technical assumptions, the final TD error is actually exactly on the order of $O(\alpha)$. This justifies the tightness of the existing upper bounds for the final TD error [6, 45] up to a constant factor. From the above corollary, we can also see the trade-off between the convergence rate and the steady state error. Clearly, the convergence rate in (27) also depends on the initial distribution $p^0$ and the mixing rate of the underlying Markov jump parameter $\{z^k\}$ (which is denoted as $\tilde{\rho}$). If the initial distribution is the stationary distribution, i.e. $p^0 = p^\infty$, the input to the LTI dynamical system (20) is just a constant for all $k$ and then we will be able to obtain an exact formula similar to (16). However, for a general initial distribution $p^0$, the mixing rate $\tilde{\rho}$ matters more and may affect the overall convergence rate. One resultant guideline for algorithm design is that increasing $\alpha$ may not increase the convergence rate when the mixing rate $\tilde{\rho}$ dominates the convergence process. When $\alpha$ becomes smaller and smaller, eventually $\sigma(\mathcal{H})$ is going to become the dominating term and the mixing rate does not affect the convergence rate any more. Similar to the IID case, for sufficiently small $\alpha$, it seems possible to obtain alternative upper bounds in the form of $\mathbb{E}\|\theta^k - \theta^*\|^2 \le \delta^\infty + O(\alpha) + C_0(\sigma(\mathcal{H}_{22}) + \varepsilon)^k$ where $\sigma(\mathcal{H}_{22}) \approx 1 + 2\operatorname{real}(\lambda_{\max\,\mathrm{real}}(\bar{A}))\alpha$. Such modifications are not pursued in this paper.

**Algorithm design.** Here we make a remark on how our proposed MJLS framework can be further extended to provide clues for designing fast TD learning. When $\alpha$ (or even other hyperparameters including momentum term) is changing with time, we can still obtain expressions of $\operatorname{vec}(Q^k)$ and $q^k$ in an iterative form. However, both $\mathcal{H}$ and $\mathcal{G}$ depend on $k$ now. Then given a fixed time budget $T$, in theory it is possible to minimize the mean square estimation error at $T$ subject to some optimization constraints in the form of a time-varying iteration $\begin{bmatrix} q^{k+1} \\ \operatorname{vec}(Q^{k+1}) \end{bmatrix} = \mathcal{H}(k) \begin{bmatrix} q^k \\ \operatorname{vec}(Q^k) \end{bmatrix} + \mathcal{G}(k)u^k$. One may use this control-oriented optimization formulation to gain some theoretical insights on how to choose hyperparameters adaptively for fast TD learning. Clearly, solving such an optimization problem requires knowing the underlying Markov model. However, this type of theoretical study may lead to new hyperparameter tuning heuristics that do not require the model information.

## Footnotes

[1] This standard assumption is typically related to the projected Bellman equation and can always be enforced by a shifting argument. More explanations are provided in Remark 1.

[2]One such condition is that $\lambda_{\max}(I_{n_{\xi}^2} + \alpha(I \otimes \bar{A} + \bar{A} \otimes I))$ is a semisimple eigenvalue.

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
