[Supplementary Material]

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

# Supplementary Material

## A   More facts about LTI systems

### A.1   Tightness of the spectral radius stability condition

Technically speaking, the condition $\sigma(\mathcal{H}) < 1$ is necessary and sufficient for the asymptotic stability of the LTI system (1). Since exponential stability and asymptotic stability are equivalent notions of stability for LTI systems, the condition $\sigma(\mathcal{H}) < 1$ is also necessary and sufficient for the exponential stability of (1). See Theorem 8.3 and Theorem 8.4 in [27] for formal statements of these facts. We will give a more intuitive explanation here. Specifically, we look at the behaviors of the matrix power term $(\mathcal{H})^k x^0$. This is the homogeneous state response of (1). There are three possible behaviors for this term.

1. When $\mathcal{H}$ is Schur stable (or equivalently $\sigma(\mathcal{H}) < 1$), the term $(\mathcal{H})^k$ converges to a zero matrix and $(\mathcal{H})^k x^0 \to 0$ for any arbitrary $x^0$. The convergence rate is linear and is completely specified by the spectral radius $\sigma(\mathcal{H})$.

2. When $\sigma(\mathcal{H}) \leq 1$ and all the Jordan blocks corresponding to eigenvalues with magnitude equal to 1 are $1 \times 1$, $(\mathcal{H})^k$ remains bounded for any $k$. This is the so-called marginal stability case where $(\mathcal{H})^k x^0$ remains bounded but may not converge to 0.

3. For all other cases, $(\mathcal{H})^k$ is unbounded and there exists $x^0$ such that $(\mathcal{H})^k x^0 \to \infty$.

See Section 7.2 in [27] for a detailed explanation of the above fact. Consequently, we can only guarantee $(\mathcal{H})^k x^0$ to converge for all $x^0$ when $\sigma(\mathcal{H}) < 1$. Therefore, the condition $\sigma(\mathcal{H}) < 1$ is a tight condition for the stability of the LTI system (1). As mentioned above, when $\sigma(\mathcal{H}) < 1$, $(\mathcal{H})^k x^0$ converges at a linear rate completely determined by $\sigma(\mathcal{H})$. Technically speaking, the convergence rate is either equal to $\sigma(\mathcal{H}) + \varepsilon$ for some arbitrary small positive $\varepsilon$ or just equal to $\sigma(\mathcal{H})$ itself. Now we provide a detailed discussion on this convergence rate.

### A.2   Convergence rate of the matrix power

Notice we have $(\mathcal{H})^k x^0 = \rho^k \left( \rho^{-1} \mathcal{H} \right)^k x^0$. As long as $\left( \rho^{-1} \mathcal{H} \right)^k x^0$ stays bounded for any $x^0$, the term $(\mathcal{H})^k x^0$ will converge at the linear rate $\rho$.

From Section 7.2 in [27], we can almost directly see how to determine the convergence rate of $(\mathcal{H})^k x^0$.

1. When $\sigma(\mathcal{H}) < 1$ and all the Jordan blocks corresponding to eigenvalues with magnitude equal to $\sigma(\mathcal{H})$ are $1 \times 1$, we can choose $\rho = \sigma(\mathcal{H})$ and show $\sigma(\rho^{-1}\mathcal{H}) \leq 1$ and all the Jordan blocks corresponding to eigenvalues (of $\rho^{-1}\sigma(\mathcal{H})$) with magnitude equal to 1 are $1 \times 1$. Then $(\rho^{-1}\mathcal{H})^k x^0$ remains bounded for all $x^0$ and hence $(\mathcal{H})^k x^0 = \rho^k \left( \rho^{-1}\mathcal{H} \right)^k x^0$ converges at a linear rate $\rho = \sigma(\mathcal{H})$.

2. When $\sigma(\mathcal{H}) < 1$ and some of the Jordan blocks corresponding to eigenvalues with magnitude equal to $\sigma(\mathcal{H})$ are not $1 \times 1$, we need to choose $\rho = \sigma(\mathcal{H}) + \varepsilon$ for some arbitrary small $\varepsilon > 0$. Then $\sigma(\rho^{-1}\mathcal{H}) < 1$ and $(\rho^{-1}\mathcal{H})^k x^0$ converges to 0 for all $x^0$. Consequently, $(\mathcal{H})^k x^0 = \rho^k \left( \rho^{-1}\mathcal{H} \right)^k x^0$ converges at a linear rate $\rho = \sigma(\mathcal{H}) + \varepsilon$.

In this paper, for simplicity we do not want to further look at the Jordan decomposition structure of $\mathcal{H}$ and hence we always set the rate as $\rho = \sigma(\mathcal{H}) + \varepsilon$. One can also use the relationship between spectral radius and other matrix norms to obtain the above convergence rate. See Section 2.2 in [39] for a detailed discussion. The arbitrarily small number $\varepsilon$ also appears in the argument there.

## B   More discussions about Markov jump linear systems

First, we verify that (7) and (8) are equivalent to the LTI model (9).

Rewriting (7) as an LTI model is quite trivial. Rewriting (8) as an LTI system requires applying the vectorization operation to obtain the following formula,

$$
\begin{bmatrix} \text{vec}(Q_1^{k+1}) \\ \vdots \\ \text{vec}(Q_n^{k+1}) \end{bmatrix} = \begin{bmatrix} p_{11}H_1 \otimes H_1 & \dots & p_{n1}H_n \otimes H_n \\ \vdots & \ddots & \vdots \\ p_{1n}H_1 \otimes H_1 & \dots & p_{nn}H_n \otimes H_n \end{bmatrix} \begin{bmatrix} \text{vec}(Q_1^k) \\ \vdots \\ \text{vec}(Q_n^k) \end{bmatrix} + \begin{bmatrix} p_{11}p_1^k I_{n_\xi^2} & \dots & p_{n1}p_n^k I_{n_\xi^2} \\ \vdots & \ddots & \vdots \\ p_{1n}p_1^k I_{n_\xi^2} & \dots & p_{nn}p_n^k I_{n_\xi^2} \end{bmatrix} \begin{bmatrix} G_1 \otimes G_1 \\ \vdots \\ G_n \otimes G_n \end{bmatrix}
$$

$$
+ \begin{bmatrix} p_{11}(G_1 \otimes H_1 + H_1 \otimes G_1) & \dots & p_{n1}(G_n \otimes H_n + H_n \otimes G_n) \\ \vdots & \ddots & \vdots \\ p_{1n}(G_1 \otimes H_1 + H_1 \otimes G_1) & \dots & p_{nn}(G_n \otimes H_n + H_n \otimes G_n) \end{bmatrix} \begin{bmatrix} q_1^k \\ \vdots \\ q_n^k \end{bmatrix}.
$$
(B.1)

Then we can augment the update rules for $q^k$ and $\text{vec}(Q^k)$ to obtain the desired LTI model for $(q^k, \text{vec}(Q^k))$.

Now we briefly review how to analyze $q^k$ and $Q^k$ using the LTI model (9). We denote $\mathcal{H} = \begin{bmatrix} \mathcal{H}_{11} & 0 \\ \mathcal{H}_{21} & \mathcal{H}_{22} \end{bmatrix}$.

First, we will have the following closed-form formula for computing $(q^k, \text{vec}(Q^k))$:

$$
\begin{bmatrix} q^k \\ \text{vec}(Q^k) \end{bmatrix} = \left( \begin{bmatrix} \mathcal{H}_{11} & 0 \\ \mathcal{H}_{21} & \mathcal{H}_{22} \end{bmatrix} \right)^k \begin{bmatrix} q^0 \\ \text{vec}(Q^0) \end{bmatrix} + \sum_{t=0}^{k-1} \left( \begin{bmatrix} \mathcal{H}_{11} & 0 \\ \mathcal{H}_{21} & \mathcal{H}_{22} \end{bmatrix} \right)^{k-1-t} \begin{bmatrix} u_q^t \\ u_Q^t \end{bmatrix}.
$$
(B.2)

The first term on the right side of the above equation will be guaranteed to converge to 0 if we have the stability condition $\sigma(\mathcal{H}) < 1$. As discussed in A.1, the stability condition $\sigma(\mathcal{H}) < 1$ is fairly tight. Based on Proposition 3.6 in [14], we know that $\mathcal{H}$ is Schur stable if and only if $\mathcal{H}_{22}$ is Schur stable. Therefore, the needed stability condition is $\sigma(\mathcal{H}_{22}) < 1$. Under this condition, if we have $p_i^k \to p_i^\infty$, then Statement 2 in Proposition 3 can be used to show

$$
u_q^\infty = \lim_{k \to \infty} u_q^k = \begin{bmatrix} p_{11}G_1 & \dots & p_{n1}G_n \\ \vdots & \ddots & \vdots \\ p_{1n}G_1 & \dots & p_{nn}G_n \end{bmatrix} \begin{bmatrix} p_1^\infty I_{n_\xi} \\ \vdots \\ p_n^\infty I_{n_\xi} \end{bmatrix},
$$

$$
u_Q^\infty = \lim_{k \to \infty} u_Q^k = \begin{bmatrix} p_{11}G_1 \otimes G_1 & \dots & p_{n1}G_n \otimes G_n \\ \vdots & \ddots & \vdots \\ p_{1n}G_1 \otimes G_1 & \dots & p_{nn}G_n \otimes G_n \end{bmatrix} \begin{bmatrix} p_1^\infty I_{n_\xi^2} \\ \vdots \\ p_n^\infty I_{n_\xi^2} \end{bmatrix},
$$
(B.3)

$$
\begin{bmatrix} q^\infty \\ \text{vec}(Q^\infty) \end{bmatrix} = \lim_{k \to \infty} \begin{bmatrix} q^k \\ \text{vec}(Q^k) \end{bmatrix} = (I - \mathcal{H})^{-1} \begin{bmatrix} u_q^\infty \\ u_Q^\infty \end{bmatrix}.
$$

Finally, if $\|p^k - p^\infty\| \le C_p \tilde{\rho}^k$, then there exists a constant $C$ such that the following inequality holds.

$$
\| \begin{bmatrix} u_q^k \\ u_Q^k \end{bmatrix} - \begin{bmatrix} u_q^\infty \\ u_Q^\infty \end{bmatrix} \| \le C\tilde{\rho}^k.
$$

If we know $\sigma(\mathcal{H}_{22}) < 1$, then we can directly apply Proposition 3 to obtain the following linear convergence result:

$$
\| \begin{bmatrix} q^k \\ \text{vec}(Q^k) \end{bmatrix} - \begin{bmatrix} q^\infty \\ \text{vec}(Q^\infty) \end{bmatrix} \| \le C_0 \max\{\sigma(\mathcal{H}) + \varepsilon, \tilde{\rho}\}^k,
$$

where $C_0$ is some constant and $\varepsilon$ is an arbitrarily small positive number. We can see that the convergence rates of $\text{vec}(Q^k)$ and $q^k$ depend on both $\sigma(\mathcal{H})$ and the mixing rate of the underlying Markov jump parameter $\{z^k\}$ (which is denoted as $\tilde{\rho}$).

Therefore, when the underlying Markov chain $\{z^k\}$ is aperiodic and irreducible, the mean and covariance of the MJLS just converges to the steady state values at a linear rate specified by $\max\{\sigma(\mathcal{H}) + \varepsilon, \tilde{\rho}\}$. This is a powerful result that can be potentially applied to more general stochastic approximation schemes other than (5). We also want to mention that there are other proofs for the convergence of $(q^k, Q^k)$. See Proposition 3.36 in [14] for an alternative proof. Here, our result is a little bit stronger than Proposition 3.36 in [14] since we also specify the convergence rate of $(q^k, Q^k)$.

Finally, it is worth mentioning that under the assumption $\sigma(\mathcal{H}_{22}) < 1$, one can further prove $\{Q^k\}$ converges to a stationary process in some sense. This is a stronger result. Specifically, Proposition 3.37 In [14] shows that the MJLS is "asymptotically wide sense stationary" under the assumption $\sigma(\mathcal{H}_{22}) < 1$. We are not that interested in the correlation between the updates at different steps since our main purpose is analyzing TD learning. Hence we will skip a detailed discussion on this topic. See Chapter 3.4 in [14] for a thorough treatment.

# C Detailed proofs

## C.1 Detailed proofs of Propositions 1, 2, and 3

We believe that all the statements in Propositions 1, 2, and 3 are known in the controls field. Since we are not able to find a reference to exactly match the statements, we provide a proof here for completeness.

We will need the following lemma.

**Lemma C.1.** *Consider the LTI model* (1). *Suppose* $\sigma(\mathcal{H}) < 1$. *We set* $\rho = \sigma(\mathcal{H}) + \varepsilon$ *where* $\varepsilon$ *is an arbitrary small positive number. Then there exists a positive definite matrix* $V$ *and a positive constant* $\gamma$ *s.t. the following inequality holds for all* $k$,

$$(x^{k+1})^{\mathsf{T}}Vx^{k+1} \leq \rho^2(x^k)^{\mathsf{T}}Vx^k + \gamma\|u^k\|^2. \tag{C.1}$$

*Proof.* We know $\rho^{-1}\mathcal{H}$ is Schur stable. Based on Theorem 8.4 in [27], there exists a positive definite matrix $V$ such that

$$\rho^{-2}\mathcal{H}^{\mathsf{T}}V\mathcal{H} - V < 0,$$

where the matrix inequality holds in the negative definite sense. The above condition is actually equivalent to $\mathcal{H}^{\mathsf{T}}V\mathcal{H} - \rho^2 V < 0$. Choose the matrix $V$ that satisfies $\mathcal{H}^{\mathsf{T}}V\mathcal{H} - \rho^2 V < 0$. Then there exists a sufficiently large $\gamma$ such that $\mathcal{G}^{\mathsf{T}}V\mathcal{G} - \gamma I < 0$ and $\mathcal{H}^{\mathsf{T}}V\mathcal{H} - \rho^2\mathcal{H} - \mathcal{H}^{\mathsf{T}}V\mathcal{G}(\mathcal{G}^{\mathsf{T}}V\mathcal{G} - \gamma I)^{-1}\mathcal{G}^{\mathsf{T}}V\mathcal{H} < 0$. By Schur complement lemma, this is equivalent to

$$\begin{bmatrix} \mathcal{H}^{\mathsf{T}}V\mathcal{H} - \rho^2 V & \mathcal{H}^{\mathsf{T}}V\mathcal{G} \\ \mathcal{G}^{\mathsf{T}}V\mathcal{H} & \mathcal{G}^{\mathsf{T}}V\mathcal{G} \end{bmatrix} + \begin{bmatrix} 0 & 0 \\ 0 & -\gamma I \end{bmatrix} < 0.$$

Now we left and right multiply the right side of the above matrix inequality with $\begin{bmatrix} (x^k)^{\mathsf{T}} & (u^k)^{\mathsf{T}} \end{bmatrix}$ and $\begin{bmatrix} (x^k)^{\mathsf{T}} & (u^k)^{\mathsf{T}} \end{bmatrix}^{\mathsf{T}}$. This leads to

$$\begin{bmatrix} x^k \\ u^k \end{bmatrix}^{\mathsf{T}} \begin{bmatrix} \mathcal{H}^{\mathsf{T}}V\mathcal{H} - \rho^2 V & \mathcal{H}^{\mathsf{T}}V\mathcal{G} \\ \mathcal{G}^{\mathsf{T}}V\mathcal{H} & \mathcal{G}^{\mathsf{T}}V\mathcal{G} \end{bmatrix} \begin{bmatrix} x^k \\ u^k \end{bmatrix} + \begin{bmatrix} x^k \\ u^k \end{bmatrix}^{\mathsf{T}} \begin{bmatrix} 0 & 0 \\ 0 & -\gamma I \end{bmatrix} \begin{bmatrix} x^k \\ u^k \end{bmatrix} \leq 0.$$

One can verify that the first term on the left side of the above inequality is just equal to $(x^{k+1})^{\mathsf{T}}Vx^{k+1} - \rho^2(x^k)^{\mathsf{T}}Vx^k$ as follows

$$(x^{k+1})^{\mathsf{T}}Vx^{k+1} - \rho^2(x^k)^{\mathsf{T}}Vx^k = (\mathcal{H}x^k + \mathcal{G}u^k)^{\mathsf{T}}V(\mathcal{H}x^k + \mathcal{G}u^k) - \rho^2(x^k)^{\mathsf{T}}Vx^k$$

$$= \begin{bmatrix} x^k \\ u^k \end{bmatrix}^{\mathsf{T}} \begin{bmatrix} \mathcal{H}^{\mathsf{T}}V\mathcal{H} - \rho^2 V & \mathcal{H}^{\mathsf{T}}V\mathcal{G} \\ \mathcal{G}^{\mathsf{T}}V\mathcal{H} & \mathcal{G}^{\mathsf{T}}V\mathcal{G} \end{bmatrix} \begin{bmatrix} x^k \\ u^k \end{bmatrix}.$$

Therefore, we have $(x^{k+1})^{\mathsf{T}}Vx^{k+1} - \rho^2(x^k)^{\mathsf{T}}Vx^k - \gamma\|u^k\|^2 \leq 0$, which is equivalent to (C.1). $\square$

Now we are ready to prove Proposition 1. Since $\sigma(\mathcal{H}) < 1$, $x^\infty$ can still be well defined as $x^\infty = (I - \mathcal{H})^{-1}\mathcal{G}u$. Notice we have not shown the existence of $\lim_{k\to\infty} x^k$ at this point. We will show $\lim_{k\to\infty} x^k$ exists and is equal to $x^\infty$. Applying the relation $(I - \mathcal{H})x^\infty = \mathcal{G}u^\infty$, we still have

$$x^{k+1} - x^\infty = \mathcal{H}(x^k - x^\infty) + \mathcal{G}(u^k - u^\infty).$$

By Lemma C.1, there exists a positive definite matrix $V$ and a positive number $\gamma$ such that

$$(x^{k+1} - x^\infty)^{\mathsf{T}}V(x^{k+1} - x^\infty) \leq \rho^2(x^k - x^\infty)^{\mathsf{T}}V(x^k - x^\infty) + \gamma\|u^k - u^\infty\|^2, \tag{C.2}$$

where $\rho = \sigma(\mathcal{H}) + \varepsilon < 1$. First we show $(x^k - x^\infty)^{\mathsf{T}}V(x^k - x^\infty)$ is bounded for all $k$ and then we apply $\limsup$ to the above inequality. Clearly there exists a constant $U$ such that $\|u^k - u^\infty\|^2 \leq U$ for all $k$. Then we have

$$(x^k - x^\infty)^{\mathsf{T}}V(x^k - x^\infty) \leq \rho^{2k}(x^0 - x^\infty)^{\mathsf{T}}V(x^0 - x^\infty) + \sum_{t=0}^{\infty}\rho^{2t}\gamma U \leq (x^0 - x^\infty)^{\mathsf{T}}V(x^0 - x^\infty) + \frac{\gamma U}{1 - \rho^2}.$$

Therefore, $\limsup_{k\to\infty}(x^k - x^\infty)^{\mathsf{T}}V(x^k - x^\infty)$ is finite. Now we take $\limsup$ on both sides of (C.2) and will immediately be able to show $\limsup_{k\to\infty}(x^k - x^\infty)^{\mathsf{T}}V(x^k - x^\infty) = 0$. Since $V$ is positive definite, we have $x^k \to x^\infty$, which is the desired conclusion.

Next, we prove Proposition 2. If $u^k = u$ for all $k$, we have $\sum_{t=0}^{k-1}(\mathcal{H})^{k-1-t}\mathcal{G}u^t = \sum_{t=0}^{k-1}(\mathcal{H})^t\mathcal{G}u$. When $\sigma(\mathcal{H}) < 1$, we have $(\mathcal{H})^k \to 0$ and $\sum_{k=0}^{\infty}(\mathcal{H})^k = (I - \mathcal{H})^{-1}$. Hence we have $x^\infty = \lim_{k\to\infty} x^k =$

$(I - \mathcal{H})^{-1}\mathcal{G}u$. Clearly, $(I - \mathcal{H})$ is nonsingular due to the fact $\sigma(\mathcal{H}) < 1$. Therefore, we have $(I - \mathcal{H})x^\infty = \mathcal{G}u^\infty = \mathcal{G}u$. Now it is straightforward to show

$$x^{k+1} - x^\infty = \mathcal{H}x^k + \mathcal{G}u^k - x^\infty = \mathcal{H}(x^k - x^\infty) + \mathcal{G}u^k - (I - \mathcal{H})x^\infty = \mathcal{H}(x^k - x^\infty),$$

which directly leads to the desired conclusion $x^k = x^\infty + (\mathcal{H})^k(x^0 - x^\infty)$.

Finally, we will still use (C.2) to prove Proposition 3. It is assumed that the arbitrary small $\varepsilon$ is chosen in a way that $\varepsilon + \sigma(\mathcal{H}) \neq \tilde{\rho}$ since one can always decrease $\varepsilon$ by a tiny bit. Then iterating (C.2) leads to

$$(x^k - x^\infty)^\mathsf{T}V(x^k - x^\infty) \leq \rho^{2k}(x^0 - x^\infty)^\mathsf{T}V(x^0 - x^\infty) + \gamma\sum_{t=0}^{k-1}\rho^{2(k-1-t)}\|u^t - u^\infty\|^2$$

$$\leq \rho^{2k}(x^0 - x^\infty)^\mathsf{T}V(x^0 - x^\infty) + C^2\gamma\sum_{t=0}^{k-1}\rho^{2(k-1-t)}\tilde{\rho}^{2t}$$

$$= \rho^{2k}(x^0 - x^\infty)^\mathsf{T}V(x^0 - x^\infty) + \left(\frac{C^2\gamma}{\rho^2 - \tilde{\rho}^2}\right)(\rho^{2k} - \tilde{\rho}^{2k}).$$

Obviously the right side of the above inequality is on the order of $O\left((\max\{\rho, \tilde{\rho}\})^{2k}\right)$. Hence we have

$$\|x^k - x^\infty\|^2 \leq \frac{1}{\lambda_{\min}(V)}(x^k - x^\infty)^\mathsf{T}V(x^k - x^\infty) = O\left((\max\{\rho, \tilde{\rho}\})^{2k}\right),$$

which leads to $\|x^k - x^\infty\| = O\left((\max\{\rho, \tilde{\rho}\})^k\right)$. This completes the proof of this proposition.

An interesting thing is that when $\rho = \tilde{\rho}$, the convergence rate is actually on the order of $O(k\rho^k)$. Specifically, we have

$$(x^k - x^\infty)^\mathsf{T}V(x^k - x^\infty) \leq \rho^{2k}(x^0 - x^\infty)^\mathsf{T}V(x^0 - x^\infty) + C^2\gamma k\rho^{2(k-1)}.$$

Of course this rate is always bounded above by $O((\varepsilon + \rho)^k)$. In addition, if it happens $\varepsilon + \sigma(\mathcal{H}) = \tilde{\rho}$, one can always decrease $\varepsilon$ by a tiny bit and the convergence rate becomes linear again.

## C.2 A detailed proof for Theorem 1

The underlying probability space is denoted by $(\Omega, \mathcal{F}, \mathbb{P})$. We denote by $\mathcal{F}_k$ the $\sigma$-algebra generated by $(z^0, z^1, \ldots, z^k)$. Clearly, $z^k$ is $\mathcal{F}_k$-adapted and we obtain a filtered probability space $(\Omega, \mathcal{F}, \{\mathcal{F}_k\}, \mathbb{P})$ on which the stochastic optimization method is defined.

First, we prove $\mu^{k+1} = (I + \alpha\bar{A})\mu^k$. Since $\mathbb{E}b(z^k) = \sum_{i=1}^n p_i b_i = 0$, we have

$$\mathbb{E}(\xi^{k+1}|\mathcal{F}_{k-1}) = \sum_{i=1}^n p_i\left((I + \alpha A_i)\xi^k + \alpha b_i\right) = \left(I + \alpha(\sum_{i=1}^n p_i A_i)\right)\xi^k + \alpha\sum_{i=1}^n p_i b_i = (I + \alpha\bar{A})\xi^k.$$

Taking full expectation of the above equation leads to $\mu^{k+1} = (I + \alpha\bar{A})\mu^k$.

Next, we prove $\mathbb{Q}^{k+1} = \mathbb{Q}^k + \alpha(\bar{A}\mathbb{Q}^k + \mathbb{Q}^k\bar{A}^\mathsf{T}) + \alpha^2\sum_{i=1}^n p_i(A_i\mathbb{Q}^k A_i^\mathsf{T} + 2\operatorname{sym}(A_i\mu^k b_i^\mathsf{T}) + b_i b_i^\mathsf{T})$. We can use a similar argument. We have

$$\mathbb{E}(\xi^{k+1}(\xi^{k+1})^\mathsf{T}|\mathcal{F}_{k-1})$$

$$= \sum_{i=1}^n\left(p_i((I + \alpha A_i)\xi^k + \alpha b_i)((I + \alpha A_i)\xi^k + \alpha b_i)^\mathsf{T}\right)$$

$$= \sum_{i=1}^n p_i(I + \alpha A_i)\xi^k(\xi^k)^\mathsf{T}(I + \alpha A_i)^\mathsf{T} + \sum_{i=1}^n \alpha p_i b_i(\xi^k)^\mathsf{T}(I + \alpha A_i)^\mathsf{T} + \sum_{i=1}^n \alpha p_i(I + \alpha A_i)\xi^k b_i^\mathsf{T} + \alpha^2\sum_{i=1}^n p_i b_i b_i^\mathsf{T}.$$

Taking full expectation and applying the fact $\sum_{i=1}^n p_i b_i = 0$ leads to

$$\mathbb{Q}^{k+1} = \sum_{i=1}^n p_i(I + \alpha A_i)\mathbb{Q}^k(I + \alpha A_i)^\mathsf{T} + \alpha^2\sum_{i=1}^n p_i\left(b_i(\mu^k)^\mathsf{T}A_i^\mathsf{T} + A_i\mu^k b_i^\mathsf{T} + b_i b_i^\mathsf{T}\right)$$

$$= \mathbb{Q}^k + \alpha(\bar{A}\mathbb{Q}^k + \mathbb{Q}^k\bar{A}^\mathsf{T}) + \alpha^2\sum_{i=1}^n p_i\left(A_i\mathbb{Q}^k A_i^\mathsf{T} + 2\operatorname{sym}(A_i\mu^k b_i^\mathsf{T}) + b_i b_i^\mathsf{T}\right).$$

This proves the recursive formula for $\mathbb{Q}^k$. Now we can apply the vectorization operation to this formula. For any matrices $A$, $X$, and $B$, we have $\mathrm{vec}(AXB) = (B^\mathsf{T} \otimes A)\,\mathrm{vec}(X)$. Hence we can directly show

$$\mathrm{vec}(\mathbb{Q}^{k+1}) = \mathrm{vec}(\mathbb{Q}^k) + \alpha(\mathrm{vec}(\bar{A}\mathbb{Q}^k) + \mathrm{vec}(\mathbb{Q}^k \bar{A}^\mathsf{T})) + \alpha^2 \sum_{i=1}^{n} p_i \,\mathrm{vec}\left(A_i \mathbb{Q}^k A_i^\mathsf{T} + 2\,\mathrm{sym}(A_i \mu^k b_i^\mathsf{T}) + b_i b_i^\mathsf{T}\right)$$

$$= \mathrm{vec}(\mathbb{Q}^k) + \alpha(I \otimes \bar{A} + \bar{A} \otimes I)\,\mathrm{vec}(\mathbb{Q}^k) + \alpha^2 \left(\sum_{i=1}^{n} p_i A_i \otimes A_i\right)\mathrm{vec}(\mathbb{Q}^k)$$

$$+ \alpha^2 \left(\sum_{i=1}^{n} p_i(b_i \otimes A_i + A_i \otimes b_i)\right)\mu^k + \alpha^2 \sum_{i=1}^{n} p_i(b_i \otimes b_i).$$

Therefore, we have $\mathrm{vec}(\mathbb{Q}^{k+1}) = \mathcal{H}_{22}\,\mathrm{vec}(Q^k) + \mathcal{H}_{21}\mu^k + \alpha^2 \sum_{i=1}^n p_i(b_i \otimes b_i)$ where $\mathcal{H}_{21}$ and $\mathcal{H}_{22}$ are determined by (15). Putting this together with $\mu^{k+1} = (I + \alpha\bar{A})\mu^k$ gives us the LTI model in (14). Then notice we have

$$\left(\begin{bmatrix} \mathcal{H}_{11} & 0 \\ \mathcal{H}_{21} & \mathcal{H}_{22} \end{bmatrix}\right)^t \begin{bmatrix} 0 \\ \alpha^2 \sum_{i=1}^n p_i(b_i \otimes b_i) \end{bmatrix} = \alpha^2 \begin{bmatrix} 0 \\ (\mathcal{H}_{22})^t (\sum_{i=1}^n p_i(b_i \otimes b_i)) \end{bmatrix}.$$

Recall that we have $\mathcal{H} = \begin{bmatrix} \mathcal{H}_{11} & 0 \\ \mathcal{H}_{21} & \mathcal{H}_{22} \end{bmatrix}$. We can apply Proposition 3.6 in [14] to show that $\mathcal{H}$ is Schur stable if and only if $\mathcal{H}_{22}$ is Schur stable. Therefore, a direct application of Proposition 3 will lead to (16). This completes the proof for Theorem 1.

## C.3 A detailed proof for Theorem 2

One may prove this theorem as a corollary of Proposition 3.35 in [14]. For completeness, we add more detailed calculations and present the proof in a self-contained manner. One can update $q_j^{k+1}$ as

$$q_j^{k+1} = \sum_{i=1}^{n} \mathbb{E}\left((H(z^k)\xi^k + \alpha b(z^k))\mathbf{1}_{\{z^k=i\}}\mathbf{1}_{\{z^{k+1}=j\}}\right)$$

$$= \sum_{i=1}^{n} \left(H_i \mathbb{E}(\xi^k \mathbf{1}_{\{z^k=i\}})\mathbb{P}(\mathbf{1}_{\{z^{k+1}=j\}}|\xi^k \mathbf{1}_{\{z^k=i\}}) + \alpha\mathbb{E}(b(z^k)\mathbf{1}_{\{z^k=i\}})\mathbb{P}(\mathbf{1}_{\{z^{k+1}=j\}}|\xi^k \mathbf{1}_{\{z^k=i\}})\right)$$

$$= \sum_{i=1}^{n} p_{ij}\left((I + \alpha A_i)q_i^k + \alpha p_i^k b_i\right).$$

This leads to the following update rule for $q^k$:

$$\begin{bmatrix} q_1^{k+1} \\ \vdots \\ q_n^{k+1} \end{bmatrix} = \begin{bmatrix} p_{11}(I + \alpha A_1) & \dots & p_{n1}(I + \alpha A_n) \\ \vdots & \ddots & \vdots \\ p_{1n}(I + \alpha A_1) & \dots & p_{nn}(I + \alpha A_n) \end{bmatrix}\begin{bmatrix} q_1^k \\ \vdots \\ q_n^k \end{bmatrix} + \alpha \begin{bmatrix} p_{11}p_1^k I & \dots & p_{n1}p_n^k I \\ \vdots & \ddots & \vdots \\ p_{1n}p_1^k I & \dots & p_{nn}p_n^k I \end{bmatrix}\begin{bmatrix} b_1 \\ \vdots \\ b_n \end{bmatrix},$$

which can be compactly written as $q^{k+1} = (P^\mathsf{T} \otimes I)\,\mathrm{diag}(I + \alpha A_i)q^k + \alpha((P^\mathsf{T}\,\mathrm{diag}(p_i^k)) \otimes I_{n_\xi})b$, where $b$ is the augmented vector

$$b = \begin{bmatrix} b_1 \\ b_2 \\ \vdots \\ b_n \end{bmatrix}.$$

This proves $q^{k+1} = \mathcal{H}_{11}q^k + \alpha((P^\mathsf{T}\,\mathrm{diag}(p_i^k)) \otimes I_{n_\xi})b$, where $\mathcal{H}_{11}$ is given by (21).

Next, we perform similar steps to obtain the iterative formula for $Q^k$. One can update $Q_j^{k+1}$ as

$$Q_j^{k+1} = \sum_{i=1}^{n} \mathbb{E}\left((H(z^k)\xi^k + \alpha b(z^k))(H(z^k)\xi^k + \alpha b(z^k))^\mathsf{T}\mathbf{1}_{\{z^k=i\}}\mathbf{1}_{\{z^{k+1}=j\}}\right)$$

$$= \sum_{i=1}^{n} \left(H_i \mathbb{E}(\xi^k(\xi^k)^\mathsf{T}\mathbf{1}_{\{z^k=i\}})H_i^\mathsf{T}\mathbb{P}(\mathbf{1}_{\{z^{k+1}=j\}}|\mathbf{1}_{\{z^k=i\}})\right) + \alpha \sum_{i=1}^{n} \left(H_i \mathbb{E}(\xi^k b(z^k)^\mathsf{T}\mathbf{1}_{\{z^k=i\}})\mathbb{P}(\mathbf{1}_{\{z^{k+1}=j\}}|\mathbf{1}_{\{z^k=i\}})\right)$$

$$+ \alpha \sum_{i=1}^{n} \left(\mathbb{E}(b(z^k)(\xi^k)^\mathsf{T}\mathbf{1}_{\{z^k=i\}})H_i^\mathsf{T}\mathbb{P}(\mathbf{1}_{\{z^{k+1}=j\}}|\mathbf{1}_{\{z^k=i\}})\right) + \alpha^2 \sum_{i=1}^{j} \mathbb{E}(b(z^k)b(z^k)^\mathsf{T}\mathbf{1}_{\{z^k=i\}})\mathbb{P}(\mathbf{1}_{\{z^{k+1}=j\}}|\mathbf{1}_{\{z^k=i\}})$$

$$= \sum_{i=1}^{n} p_{ij}\left(H_i Q_i^k H_i^\mathsf{T} + 2\alpha\,\mathrm{sym}(H_i q_i^k b_i^\mathsf{T}) + \alpha^2 p_i^k b_i b_i^\mathsf{T}\right).$$

Now we can apply the vectorization operation to obtain the following equation

$$\text{vec}(Q_j^{k+1}) = \sum_{i=1}^{n} p_{ij} \left( (H_i \otimes H_i) \text{vec}(Q_i^k) + \alpha(b_i \otimes H_i + H_i \otimes b_i)q_i^k + \alpha^2 p_i^k b_i \otimes b_i \right),$$

which is equivalent to

$$
\begin{bmatrix} \text{vec}(Q_1^{k+1}) \\ \vdots \\ \text{vec}(Q_n^{k+1}) \end{bmatrix}
=
\begin{bmatrix} p_{11}H_1 \otimes H_1 & \dots & p_{n1}H_n \otimes H_n \\ \vdots & \ddots & \vdots \\ p_{1n}H_1 \otimes H_1 & \dots & p_{nn}H_n \otimes H_n \end{bmatrix}
\begin{bmatrix} \text{vec}(Q_1^k) \\ \vdots \\ \text{vec}(Q_n^k) \end{bmatrix}
+ \alpha^2
\begin{bmatrix} p_{11}p_1^k I_{n_\xi} & \dots & p_{n1}p_n^k I_{n_\xi^2} \\ \vdots & \ddots & \vdots \\ p_{1n}p_1^k I_{n_\xi^2} & \dots & p_{nn}p_n^k I_{n_\xi^2} \end{bmatrix}
\begin{bmatrix} b_1 \otimes b_1 \\ \vdots \\ b_n \otimes b_n \end{bmatrix}
$$

$$
+ \alpha
\begin{bmatrix} p_{11}(b_1 \otimes H_1 + H_1 \otimes b_1) & \dots & p_{n1}(b_n \otimes H_n + H_n \otimes b_n) \\ \vdots & \ddots & \vdots \\ p_{1n}(b_1 \otimes H_1 + H_1 \otimes b_1) & \dots & p_{nn}(b_n \otimes H_n + H_n \otimes b_n) \end{bmatrix}
\begin{bmatrix} q_1^k \\ \vdots \\ q_n^k \end{bmatrix}.
$$

$$(\text{C.3})$$

We can compactly rewrite the above equation as $\text{vec}(Q^{k+1}) = \mathcal{H}_{22} \text{vec}(Q^k) + \mathcal{H}_{21}q^k + \alpha^2 \text{diag}(p_i^k)(P^\mathsf{T} \otimes I_{n_\xi^2})\hat{B}$, where $\mathcal{H}_{22}$ and $\mathcal{H}_{21}$ are given by (21), and $\hat{B} = \begin{bmatrix} (b_1 \otimes b_1)^\mathsf{T} & \dots & (b_n \otimes b_n)^\mathsf{T} \end{bmatrix}^\mathsf{T}$. Putting the recursion formulas for $q^k$ and $\text{vec}(Q^k)$ together leads to the desired state-space model (20). The rest of the theorem statement follows from direct applications of Equation (2).

# D   Details for perturbation analysis under the Markov assumption

The perturbation analysis in Section 5 relies on a few technical lemmas from matrix perturbation theory. We provide more details here. We will use the following fact.

**Proposition D.1.** *Suppose $\lambda$ is a simple eigenvalue of $A$ with left eigenvector $y$ and right eigenvector $x$. Suppose $B$ and $A \otimes I_m$ have the same dimension. Let $c$ be an eigenvalue of the $m \times m$ matrix $(y \otimes I_m)B(x \otimes I_m)$. Then the matrix $(A \otimes I_m) + \alpha B$ has an eigenvalue yielding the first-order expansion $\lambda + c\alpha + O(\alpha^2)$ for small $\alpha$.*

The above proposition is a special case of Theorem 2.1 in [42]. See the remark placed after Theorem 2.1 in [42] for explanations. Now we can directly apply the above proposition to analyze the spectral radius of $\mathcal{H}_{11}$ and $\mathcal{H}_{22}$. First recall that $\mathcal{H}_{11} = (P^\mathsf{T} \otimes I_{n_\xi}) \text{diag}(I_{n_\xi} + \alpha A_i) = P^\mathsf{T} \otimes I_{n_\xi} + \alpha(P^\mathsf{T} \otimes I_{n_\xi}) \text{diag}(A_i)$. Based on the ergodicity assumption on $\{z^k\}$, 1 is a simple eigenvalue of $P^\mathsf{T}$ with left eigenvector $y = \begin{bmatrix} 1 & 1 & \dots & 1 \end{bmatrix}$ and right eigenvector $p^\infty$ which is the unique stationary distribution of $\{z^k\}$. Since we have $(y \otimes I_{n_\xi})(P^\mathsf{T} \otimes I_{n_\xi}) \text{diag}(A_i)(p^\infty \otimes I_{n_\xi}) = \sum_{i=1}^{n} p_i^\infty A_i = \bar{A}$, we can directly apply the above proposition to show

$$\lambda_{\max}(\mathcal{H}_{11}) = 1 + \lambda_{\max\,\text{real}}(\bar{A})\alpha + O(\alpha^2). \tag{D.1}$$

Therefore, we have

$$\sigma(\mathcal{H}_{11}) = \sqrt{(1 + \alpha\,\text{real}(\lambda_{\max\,\text{real}}(\bar{A})))^2 + (\text{imag}(\lambda_{\max\,\text{real}}(\bar{A})))^2} \approx 1 + \text{real}(\lambda_{\max\,\text{real}}(\bar{A}))\alpha + O(\alpha^2).$$

Next, we do a similar perturbation analysis to show $\mathcal{H}_{22} = 1 + 2\,\text{real}(\lambda_{\max\,\text{real}}(\bar{A}))\alpha + O(\alpha^2)$. Recall that we have

$$\mathcal{H}_{22} = (P^\mathsf{T} \otimes I_{n_\xi^2}) \text{diag}((I_{n_\xi} + \alpha A_i) \otimes (I_{n_\xi} + \alpha A_i))$$

$$= P^\mathsf{T} \otimes I_{n_\xi^2} + \alpha((P^\mathsf{T} \otimes I_{n_\xi^2}) \text{diag}(A_i \otimes I_{n_\xi} + I_{n_\xi} \otimes A_i)) + O(\alpha^2).$$

Since we have $(y \otimes I_{n_\xi^2})(P^\mathsf{T} \otimes I_{n_\xi^2}) \text{diag}(A_i \otimes I_{n_\xi} + I_{n_\xi} \otimes A_i)(p^\infty \otimes I_{n_\xi^2}) = \bar{A} \otimes I_{n_\xi} + I_{n_\xi} \otimes \bar{A}$, we can directly apply the above matrix perturbation proposition to show

$$\lambda_{\max}(\mathcal{H}_{22}) = 1 + 2\lambda_{\max\,\text{real}}(\bar{A})\alpha + O(\alpha^2), \tag{D.2}$$

which leads to the desired first order expansion of $\sigma(\mathcal{H}_{22})$. Another fact that we used in the above argument is that all the eigenvalues of $\bar{A} \otimes I_{n_\xi} + I_{n_\xi} \otimes \bar{A}$ are in the form of a sum of two eigenvalues of $\bar{A}$.

**A remark on the IID case.**   For the IID case, we have $\mathcal{H}_{11} = I + \alpha\bar{A}$. Hence the eigenvalues of $\mathcal{H}_{22}$ are in the form of $1 + \alpha\lambda$ where $\lambda$ is an eigenvalue of $\bar{A}$. So there is no need to even perform a perturbation analysis here. We directly have

$$\sigma(\mathcal{H}_{11}) = \sqrt{(1 + \alpha\,\text{real}(\lambda_{\max\,\text{real}}(\bar{A})))^2 + (\text{imag}(\lambda_{\max\,\text{real}}(\bar{A})))^2} \approx 1 + \text{real}(\lambda_{\max\,\text{real}}(\bar{A}))\alpha + O(\alpha^2).$$

To analyze $\sigma(\mathcal{H}_{22})$, first recall that we have $\mathcal{H}_{22} = I_{n_\xi^2} + \alpha(I \otimes \bar{A} + \bar{A} \otimes I) + \alpha^2 \sum_{i=1}^{n} p_i(A_i \otimes A_i)$. If we assume $\lambda_{\max}(I_{n_\xi^2} + \alpha(I \otimes \bar{A} + \bar{A} \otimes I))$ is a semisimple eigenvalue, then we can apply Proposition D.1 to obtain $\lambda_{\max}(\mathcal{H}_{22}) = 1 + 2\lambda_{\max\,\mathrm{real}}(\bar{A}) + O(\alpha^2)$.

## E   Connections to existing finite sample bounds on mean square errors

Most existing finite sample bounds for TD learning with a constant learning rate have the following form:

$$\mathbb{E}\|\xi^k\|^2 \leq C_0 \rho^{2k} + C_1, \tag{E.1}$$

where $C_0$ is a constant, $\rho^2$ is the convergence rate, and $C_1$ quantifies the final error level. Typically one proves $\rho^2 = 1 - c\alpha + O(\alpha^2)$ for some $c$, and $C_1 = O(\alpha)$. One most relevant result of this nature for the Markov noise model was presented as Theorem 7 in [45]. Our result justifies the tightness of the result in [45] for the following reasons.

- In [45], the constant $c$ in the rate $\rho^2$ is a constant determined by $\bar{A}$. Using our perturbation analysis, we can see eventually $c$ is going to be determined by the real part of $\lambda_{\max\,\mathrm{real}}(\bar{A})$. Actually one could modify the argument in [45] to match the constant $c$ with our perturbation analysis result by choosing a slightly better Lyapunov function based on the solution of an linear matrix inequality $\bar{A}^\mathsf{T} V + V\bar{A} + 2\rho V \preceq 0$.

- In [45], the constant $C_1$ is at the order of $O(\alpha)$ which matches the perturbation analysis result obtained in our paper. It is possible that an exact formula for the constant $C_1$ can be obtained to match the steady state mean square error $\lim_{k\to\infty} \mathrm{trace}(\mathbb{Q}^k)$ more accurately, although we have not pursued such an analysis.

- In [45], the rate $\rho$ does not depend on the mixing time property. This is consistent with our theory. Based on our theory, as $\alpha$ gets smaller, the rate $\rho$ becomes independent of the mixing rate $\tilde{\rho}$, although the constant $C_0$ still has some dependence on $\tilde{\rho}$.

- Our results further show that $\mathbb{E}\|\xi^k\|^2$ converges to an exact limit at a linear rate. Hence both upper and lower bounds for the mean square TD error are simultaneously provided.

It is worth mentioning that the bounds in the form of (E.1) capture the behaviors of TD learning quite well for small $\alpha$, but can be conservative for large $\alpha$. Our formulas are exact for all $\alpha$. The generality comes at the price of loosing some interpretability for the large learning rate region. How to interpret $\sigma(\mathcal{H}_{22})$ for larger $\alpha$ in a better way remains unclear at this moment.

## F   More discussions on jump system formulations for variants of TD(0)

Finally, we present some extra details for the jump system formulations of several TD learning algorithms other than TD(0). Specifically, all the methods that can be analyzed using the ODE method has the form $\xi^{k+1} = (I + \alpha A(z^k))\xi^k + \alpha b(z^k)$. Then taking expectation of $A(z^k)$ and $b(z^k)$ under the stationary distribution and making $\alpha$ arbitrarily small leads to the ODE $\dot{\xi} = \bar{A}\xi$. As commented in Section 3, the linear stochastic approximation scheme $\xi^{k+1} = (I + \alpha A(z^k))\xi^k + \alpha b(z^k)$ is just a MJLS. Now we give detailed references for this type of formulations for various TD learning algorithms. The detailed linear stochastic approximation form for GTD is given in Section 4 of [50]. The detailed linear stochastic approximation form for GTD2 is given in Section 5 of [49]. TDC yields a similar formulation. The double temporal difference (DTD) learning method and the average temporal difference (ATD) learning method are proposed in [38]. The ODE formulations for both DTD and ATD are presented in the supplementary material of [38], yielding straightforward jump system formulations.

It is also possible to model two time-scale methods or off-policy TD learning using the general jump system model (4). One needs to slightly modify the definitions of $\{H_i\}$ and $\{G_i\}$. Then one can immediately apply the LTI model (9) to obtain closed-form formulas for the mean square error of these methods.