[Reviews · NeurIPS 2019]

Reviewer 1



*** Update: The authors have addressed my comments. But I also agree certain points brought by Reviewer 4 (e.g. the writing can be more focused on TD), so I decided to keep the same score. *** I like the linear system perspective on the update rule of temporal difference learning. Although similar ideas have been applied in the background in proofs here and there, it is good to have a comprehensive treatment based on control theory. Overall I like the perspective provided by paper. However, I'm also worried that, while one can definitely formulate that as a linear system and then apply known results in control theory. It doesn't seem to provide strong insights to temporal difference learning per se, if it introduces additional warranted assumptions to characterize the system behaviors. Below I list some concerns about certain technical points which I wish the authors can clarify. 1. Assumption on E[b(z^k)] = 0. I am wondering whether this is assumption is reasonable in practice. For example, is it true for the b(z^k) of the classic rule (13)? It seems that it depends on the distribution that determines \theta^* and whether the problem is discounted. Also, do we need still this assumption in Theorem 2? It is not clear, though it seems to be according to line 259. 2. The stability of linear systems. The authors provide analyses based on perturbation to show that the system can be stable when the step size is small enough. Nonetheless, at the end, the argument still hinge on \bar{A} is Hurwitz. How can this be ensured?. For example, under what assumption, can this be true for the simple update rule in (13)? Minor questions: Why is there a minus sign in line 213? Is it due to the Hurwtiz assumption? Typos: In (9) shouldn't it be q_i^k? A parenthesis is missing in line 142 line 167, x^k -> s^k

Reviewer 2



The paper considers TD algorithms with linear function approximations. The key observation is that the learning dynamics of weights in a TD algorithms follow MJLS dynamics. Then the paper uses MJLS theories to analyze the dynamics of TD algorithms with linear function approximations, and then from the dynamics provides stability analysis for the learning algorithms. The method provides a new systematic way to analyze the behaviors of TD learning algorithms which are important for RL research. The technical proofs are correct, but the presentation could be improved. Some comments: - The definitions of \mu^k, q^k and Q^k are hard to find and in the paper, and they are a bit confusing. It's probably better to define them in Section 3 together with the jump dynamics (12) and provide more explanations. - Theorem 1 and 2 are basically some algebraic manipulations of the TD dynamics, and the stability and limiting behavior are the main results of the paper. Therefore, it feels like providing some formal theorems for the stability and limiting behavior might better highlight the main results. - In Theorem 1, (14) uses \mu^k but (16) uses q^k. It seems like a typo in (16). - The assumption \sum p_i b_i = 0 is desirable, but not clear if it would actually hold. It feels like this assumption is related to the representability of the linear function approximations. More discussions on this assumption is important. In particular, what is the dynamics when this expected value is not zero? What is the limiting behavior when this assumption does not hold?

Reviewer 3



Summary: - The main contribution of the paper is to write the TD update as a MJLS over an augmented parameter space with one parameter vector for each pair of states in the underlying MDP. - After presenting MJLS and the idea of the augmented parameter space, they first consider the IID case where pairs of states are chosen IID and give formulas for the expected error and its covariance. - Then they move to the Markov noise case and derive formulas for the the augmented error and covariance. Under an additional ergodicity assumption they give a convergence rate to limiting quantities. For small learning rates (not exactly clear how small in terms of problem parameters) a perturbation analysis gives an estimate of what this convergence rate is (although the value of lambda_{max real} \bar A remains unclear in terms of the parameters of the problem). Pros: - The originality of the connection between TD dynamics and MJLS is a good contribution that could increase the flow of ideas from control theory to RL. In addition, the formulation of the augmented state space seems to be a potentially useful analysis tool. - The proofs seem correct with references to the appropriate work. Section D of the supplement provides an honest analysis of how the conclusions of the analysis are very similar to Srikant and Ying [22] while being more exact but less interpretable. The quality of the paper is solid. Cons: - In my mind, the main issue with this paper is a lack of significance. The approach is slightly different, but the conclusions seem to be the same as two recent papers with finite-time analysis of TD learning (references [3] and [22] in the paper). However, the results in this paper are much harder to interpret as they are presented with cumbersome notation and the impact of the derived formulas is unclear. Moreover, the paper never brings the analysis back down from the augmented space to the error in the parameters of the actual TD algorithm, making it even more difficult to interpret the results and compare them on equal footing with related work. To me, the improvement in more exact convergence rates (but of the same order) does not provide any useful insight into the way that TD learning works since the analysis is so difficult to interpret. - In general, the clarity and focus of the exposition could be improved. The paper gets bogged down in presenting several pages about MJLS before explaining the connection to TD (which is the main contribution). The notation is cumbersome and confusing leading to long and difficult to interpret theorems with sparse analysis/exposition. Connections to variants of TD are hinted at but omitted from the paper. Clarity issues are further addressed in the "improvements" section below. - Related to the above point about significance, the results lack originality. The two recent papers with finite-time analysis of TD learning (references [3] and [22] in the paper) derive similar results, with bounds on the MSE instead of exact formulas in an augmented space on the covariance which is not quite the same thing, as stated in the paper. And while the connection to MJLS is novel in reinforcement learning, it seems to have been inspired by a similar connection made in optimization in the paper by Hu et al. (reference [15]). ----------------------------------------------------------------- Update and response to authors: Interpretability: Adding corollaries that express the results as comparable to the ones in the literature will help interpretability. Also, splitting up the theorems into smaller pieces and simplifying the notation (and maybe moving some of the results to the appendix) will clean up the presentation and make things more understandable. Moreover, as the other reviewers said, more clear explanations of the assumptions should be added. Finally, spelling out the implications of the theorems more clearly will go a long way towards making the paper more impactful. Significance: I am still a bit wary of the impact on understanding TD learning. The analysis is novel and the tight rates are nice, but the claims about informing the choice of learning rate are suspect as they rely on knowing the eigenvalues of bar(A), a matrix that will not be known in practice. Expanding on the implications of tight rates and/or lower bound on the error could help explain the significance. An example or toy experiment could help get this point across. Originality: I agree that I missed the distinction with prior work that the authors raise and I think that this MJLS perspective is a nice idea to bring to the RL community. After the rebuttal and talking to the other reviewers, I will update my score to a 6 to reflect expected changes to the paper to improve clarity and interpretability and to recognise the novelty of the MJLS perspective and tight bounds. My doubts remain about the significance, but better presentation should improve this.

[Author Response · NeurIPS 2019]

We thank all the reviewers for their careful feedback and will revise our paper accordingly. We start with a re-
sponse addressing one common point raised by Reviewer 1 and Reviewer 3 regarding how to handle the case where
$\sum_{i=1}^{n} p_i^\infty b_i \neq 0$. This case can be handled by a shifting argument if $\bar{A} := \sum_{i=1}^{n} p_i^\infty A_i$ is invertible. Notice the iteration
$\xi^{k+1} = (I + \alpha A(z^k))\xi^k + \alpha b(z^k)$ can be rewritten as $\xi^{k+1} - \tilde{\xi} = \xi^k - \tilde{\xi} + \alpha \left( A(z^k)(\xi^k - \tilde{\xi}) + A(z^k)\tilde{\xi} + b(z^k) \right)$ for any
$\tilde{\xi}$. Now we denote $\tilde{b}_i = A_i \tilde{\xi} + b_i$ and the above iteration just becomes $\xi^{k+1} - \tilde{\xi} = (I + \alpha A(z^k))(\xi^k - \tilde{\xi}) + \alpha \tilde{b}(z^k)$. When
$\bar{A}$ is invertible, we can choose $\tilde{\xi} = -(\sum_{i=1}^{n} p_i^\infty A_i)^{-1}(\sum_{i=1}^{n} p_i^\infty b_i)$ such that $\sum_{i=1}^{n} p_i^\infty \tilde{b}_i = \sum_{i=1}^{n} p_i^\infty (A_i \tilde{\xi} + b_i) = 0$.
Now we can directly apply the theory in our paper to obtain analytical formulas for $\mathbb{E}(\xi^k - \tilde{\xi})$ and $\mathbb{E}[(\xi^k - \tilde{\xi})(\xi^k - \tilde{\xi})^T]$,
which eventually lead to formulas for $\mathbb{E}\xi^k$ and $\mathbb{E}[\xi^k(\xi^k)^T]$. A key question is when $\bar{A}$ will be invertible. Typically
this can be guaranteed by some rank conditions on the feature matrix $\Phi$ whose $i$-th row is equal to $\phi(i)^T$. For TD(0),
$\bar{A}$ is Hurwitz (and hence invertible) when the discount factor is smaller than 1, $p_i^\infty$ is positive for all $i$, and $\Phi$ is full
column rank. Such a fact is presented in the classic paper "An analysis of temporal-difference learning with function
approximation" by Tsitsiklis and Van Roy. Similar facts can be found for other TD algorithms (e.g. see Assumption 2
and Appendix A in Ref [19] for DTD and ATD). The assumption that $\Phi$ is full column rank is standard and states that
any redundant features have been removed. Reviewer 1 is correct in that a discount factor is needed. In our paper, the
calculation of $\theta^*$ for TD(0) already involved such a shifting argument, and the condition $\sum_{i=1}^{n} p_i^\infty b_i = 0$ is enforced
for Equation (13) due to the fact that the projected Bellman equation and the equation $\sum_{i=1}^{n} p_i^\infty b_i = 0$ are equivalent
for TD(0). Notice $\theta^*$ only solves the projected Bellman equation and does not minimize the mean-square Bellman error.
When $\bar{A}$ is singular, we can slightly modify the input terms in (14) and (20) and directly obtain analytical formulas for
$(q^k, Q^k)$. However, there is no convergence guarantee for this case. Now we address specific reviewer comments below.

**Response to Reviewer 1:** In the above response, we have already discussed the validity of the assumption $\sum_{i=1}^{n} p_i^\infty b_i = $
$0$ for TD algorithms and how to shift terms for the case where $\sum_{i=1}^{n} p_i^\infty b_i \neq 0$. Now we discuss how to ensure the
assumption that $\bar{A}$ is Hurwitz. This is a standard assumption required even by the basic ODE approach which is used to
prove asymptotic convergence. This assumption can be guaranteed by some rank conditions on the feature matrix $\Phi$.
For example, when $\Phi$ is full column rank, $\bar{A}$ is Hurwitz for Equation (13). A reference for this is the classic paper "An
analysis of temporal-difference learning with function approximation" by Tsitsiklis and Van Roy. Similar conditions
for other TD algorithms can be found in Refs [19, 25]. We emphasize that our approach does not require any extra
assumptions compared with the existing approaches. Finally, the "-" sign in Line 213 is due to the Hurwtiz assumption.

**Response to Reviewer 3:** We thank the reviewer for the constructive suggestions on how to improve the readability of
the paper. We will revise the paper accordingly. Regarding the assumption $\mathbb{E}p_i^\infty b_i = 0$, please see our response at the
beginning of this rebuttal. We also want to mention that one way of extending our approach for the infinite sample
space is by using operator theory. In this case, we will have some infinite dimensional variants of (5) and (6). Now
the iterations on $q^k$ and $Q^k$ are described by infinite dimensional linear operators instead of finite dimensional linear
operators (which are just matrices). A rigorous treatment of such extensions requires heavy mathematical notation due
to the use of spectrum theory of linear operators. We will outline such ideas (without giving details) in our revised draft.

**Response to Reviewer 4:** We agree that the new insights on TD learning brought by our analysis should be made more
transparent. We will focus more on TD learning and improve the clarity accordingly. We do think that the reviewer has
misunderstood our paper regarding its interpretability, significance, and originality. We will revise to make the following
clarifications. Regarding **interpretability**, our results are not more difficult to interpret than the mean square error
bound in Ref [23]. The trace of the covariance matrix will immediately give us the mean square error. Consequently,
by substituting the expressions of $Q^k$ into the equation in Line 141 of our paper, we will directly get exact formulas
and related bounds for the mean square error at any step $k$. Regarding **significance**, our exact formulas do bring new
insights compared with existing sample bounds. Ref [3] requires an extra projection step to handle the Markov noise,
so now we mainly compare our results with Ref [23]. Firstly, based on Statement 2 of Theorem 2 in our paper, the
covariance matrix (or the mean square error) has an exact limit. In contrast, Ref [23] only shows that the final mean
square error is bounded above. Secondly, a fundamental question is how tight the bounds in Ref [23] are. Does there
exist an ergodic Markov chain such that the resultant final mean square error actually scales on the order $O(\alpha^m)$ for
some constant $m > 1$? Our theory states that the answer to this question is no. Our exact formulas for the convergence
rate and the final limits of $(q^k, Q^k)$ can not only provide an upper bound for the mean square error, but also directly
lead to lower bounds. This justifies the tightness of the upper bounds in Ref [23]. Thirdly, for large $\alpha$ region, our
theory states that the mixing rate of the underlying Markov chain $z^k$ poses a fundamental limitation for the convergence
rate of TD learning. Statement 3 in Theorem 2 of our paper exactly characterizes this effect, and we provided further
discussions in the last paragraph of our main paper. Such a fact is not explained by the theory in Ref [23] which focuses
on small $\alpha$ region. Our theory sheds new light on how to choose large $\alpha$ at the early phase of TD learning. Regarding
**originality**, our paper is the first that uses MJLS theory to analyze learning algorithms. Although Ref [15] presents a
jump system formulation for stochastic optimization in supervised learning, the noise model there is IID and MJLS
theory is not used there. Our paper is the first one that really bridges "Markov" jump linear system theory with learning.

[Meta-Review · NeurIPS 2019]

This paper provides an analysis of Temporal-Difference algorithms using the theory of Markov Jump Linear Systems (MJLS). Its main contributions are to establish exact dynamics for the first and second order TD moments using linear function approximation, given by Linear Time Invariant systems. Reviewers found the technical contributions of this paper to be very strong, with potentially important significance in the study of a central object in RL such as TD learning. The main point of contention is the current presentation, which is cumbersome with notation, with page-long theorem statements, and, most importantly, without sufficient discussion of how these results relate to existing work on the convergence analysis of TD learning. The AC shares these concerns. However, after careful discussion with reviewers and having read the author feedback (which does promise to improve readability), considered that the positive contributions outweight the risk of poor readibility, and recommends acceptance, urging the authors to address the concerns raised by reviewers and AC.